# An inter-comparison of the mass budget of the Arctic sea ice in CMIP6 models

Ann Keen[1], Ed Blockley[1], David A. Bailey[2], Jens Boldingh Debernard[3], Mitchell Bushuk[4],
Steve Delhaye[5], David Docquier[6], Daniel Feltham[7], François Massonnet[5], Siobhan O'Farrell[8],
Leandro Ponsoni[5], José M. Rodriguez[9], David Schroeder[7], Neil Swart[10], Takahiro Toyoda[11],
Hiroyuki Tsujino[11], Martin Vancoppenolle[12], and Klaus Wyser[6]

1: Met Office Hadley Centre, Exeter, UK.
2: National Center for Atmospheric Research (NCAR), Boulder, CO, USA.
3: Norwegian Meteorological Institute, Oslo, Norway.
4: Geophysical Fluid Dynamics Laboratory (GFDL), Princeton, NJ, USA.
5: Georges Lemaître Centre for Earth and Climate Research (TECLIM), Earth and Life Institute, Université catholique de Louvain, Louvain-la-Neuve, Belgium.
6: Rossby Centre, Swedish Meteorological and Hydrological Institute (SMHI), Norrköping, Sweden.
7: Centre for Polar Observations and Modelling (CPOM), University of Reading, Reading, UK.
8: CSIRO Oceans and Atmosphere, Aspendale, Victoria, Australia.
9: Agencia Estatal de Meteorología (AEMET), Madrid, Spain.
10: Environment and Climate Change Canada (ECCC), Canadian Centre for Climate Modelling and Analysis, Victoria, BC.
11: Meteorological Research Institute, Japan Meteorological Agency, Tsukuba, Japan.
12: Laboratoire d'Océanographie et du Climat and Institut Pierre-Simon Laplace (LOCEAN-IPSL), Paris, France.

*Correspondence to*: Ann Keen (ann.keen@metoffice.gov.uk)

**Abstract.** We compare the mass budget of the Arctic sea ice for 15 models submitted to the latest Climate Model Inter-comparison Project (CMIP6), using new diagnostics that have not been available for previous model inter-comparisons. These diagnostics allow us to look beyond the standard metrics of ice cover and thickness, to compare the processes of sea ice growth and loss in climate models in a more detailed way than has previously been possible.

For the 1960-89 multi-model mean, the dominant processes causing annual ice growth are basal growth and frazil ice formation, which both occur during the winter. The main processes by which ice is lost are basal melting, top melting and advection of ice out of the Arctic. The first two processes occur in summer, while the latter process is present all year. The sea-ice budgets for individual models are strikingly similar overall in terms of the major processes causing ice growth and loss, and in terms of the time of year during which each process is important.

However, there are also some key differences between the models, and we have found a number of relationships between model formulation and components of the ice budget that hold for all or most of the CMIP6 models considered here. The relative amounts of frazil and basal ice formation vary between the models, and the amount of frazil ice formation is strongly dependent on the value chosen for the minimum frazil ice thickness. There are also differences in the relative amounts of top and basal melting, potentially dependent on how much shortwave radiation can penetrate through the sea ice into the ocean. For models with prognostic melt ponds, the choice of scheme may affect the amount of basal growth, basal melt and top melt, and the choice of thermodynamic scheme is important in determining the amount of basal growth and top melt.

As the ice cover and mass decline during the 21st century, we see a shift in the timing of the top and basal melting in the multi-model mean, with more melt occurring earlier in the year, and less melt later in the summer. The

amount of basal growth reduces in the autumn, but it increases in the winter due to thinner sea ice over the course of the 21$^{st}$ century. Overall, extra ice loss in May-June and reduced ice growth in October-November is partially offset by reduced ice melt in August and increased ice growth in January-February. For the individual models, changes in the budget components vary considerably in terms of magnitude and timing of change. However, when the evolving budget terms are considered as a function of the changing ice state itself, behaviours common to all the models emerge, suggesting that the sea ice components of the models are fundamentally responding in a broadly consistent way to the warming climate.

It is possible that this similarity in the model budgets may represent a lack of diversity in the model physics of the CMIP6 models considered here. The development of new observational datasets for validating the budget terms would help to clarify this.

## 1. Introduction

Sea ice is a key component of the climate system, and the observed decline in Arctic ice cover provides a very visible indicator of climate change. Between 1979 and 2018, Arctic September sea ice extent has declined at a rate of nearly 13% per decade (IPCC, 2019). The ice has also thinned, and there has been a transition to a greater coverage of younger ice: the proportion of Arctic ice cover that is more than 5 years old decreased by 90% between 1979 and 2018 (Kwok, 2018; Stroeve and Notz, 2018; IPCC, 2019). The ice also moves more quickly (Olason and Notz, 2014).

Global coupled climate models are the most comprehensive tools that we have for predicting how the Arctic ice will change in the future, as they can represent a range of processes that control the seasonal cycle of ice growth and melt, and so have the potential to represent the changing nature of the ice itself. However, model projections show a wide spread in the rate of ice decline, both during the period for which we have observations, and also into the future as we move towards a seasonally ice-free Arctic (Massonnet et al. 2012; Notz and Stroeve, 2016; SIMIP community, 2020).

Some of this spread is an inevitable consequence of the internal variability of the climate system, and the uncertainty in future forcing. For example, Jahn et al. (2016) find that internal variability causes an uncertainty of 21 years in predicting the year in which the Arctic first becomes seasonally ice-free using a large (40 member) ensemble of model runs, with an additional uncertainty of 5 years due to scenario uncertainty. A similar degree of spread also occurs due to differences between the models' representation of the sea ice and other components of the climate system (Topál et al., 2020). Spread may also arise from differences in initial conditions (Hawkins and Sutton, 2009; Melia et al., 2015).

For previous model inter-comparisons (CMIP3 and CMIP5), primarily only changes in ice extent, volume (or mass), and ice motion (Tandon et al., 2018, Rampal et al., 2011) have been considered, as these quantities are readily available as model output that can be compared directly to one another, and to observational or reanalysis references. However, there is an emerging consensus that in order to understand the reasons for differences in projections of the ice state, we need to be able to look 'behind the scenes' to understand the balance of different processes that drive the evolving ice state, and how these change as the ice declines.

For the CMIP3 models, Holland et al. (2010) calculated the changes in ice mass due to melt, growth and divergence using monthly mean model values of ice thickness and velocity. They found an appreciable variation in the size and relative importance of changes in these budget components between the models as the sea ice

declines. For individual models, diagnostics may be available that allow a more comprehensive decomposition of the model budget. For example, Keen and Blockley (2018) studied changes in the volume budget of the Arctic sea ice in a CMIP5 model under a range of forcing scenarios, considering both annual and seasonal changes in the individual processes causing ice growth and loss.

For the latest generation of sea ice models (the CMIP6 models), a Sea-Ice Model Intercomparison Project (SIMIP) has been established, which has defined a comprehensive set of diagnostics allowing for the intercomparison of the mass, energy and freshwater budgets of the sea ice (Notz et al., 2016). In this study we use these new diagnostics to present a first intercomparison of the mass budget of the sea ice for 15 of the CMIP6 models. Note that this is a subset of the CMIP6 models, just including those for which the required outputs were available. We

first look at the mean mass budget for a reference period, to determine the similarities and differences between the model budgets during a period with relatively little change in the ice state. We then consider how the budgets change as the ice declines during the 21$^{st}$ century, and how changes in the budget components relate to changes in the ice state and the global temperature.

In Sect. 2 we describe the models and forcing scenarios used, and in Sect. 3 we intercompare the modelled ice

area and mass. In Sects. 4 and 5 we consider the mean mass budget during a reference period, and in Sect. 6 we investigate how the budget evolves during the 21$^{st}$ century. In Sect. 7 we summarise and discuss our results.

## 2. Models and methodology

In this study we analyse data from 15 CMIP6 models, originating from 9 different modelling centres. We also analyse data from 3 configurations of the NEMO-CICE ocean-sea ice model forced by atmospheric reanalysis, which has a similar formulation to one of the CMIP6 models used in this study (HadGEM3-GC3.1-LL). These models, and the data provided from each, are listed in Table 1. Key details about the formulation of each model are summarised in Table 2, with a brief description of each in Appendix A. For each model, the ice area and mass,

and the area weighted monthly-mean ice mass budget terms were calculated over the domain shown in Fig. 1a, covering the Arctic Ocean Basin (Central Arctic plus the Beaufort, Chukchi, East Siberian, Laptev, Kara Seas) and the Barents Sea. Unless otherwise stated, all results shown in the paper are integrals over this analysis region and so do not represent the entire Northern Hemisphere ice-covered region, especially in winter. This will have an impact on some of the values calculated. Most notably it may affect the month of the seasonal maxima and

minima for some quantities, but as values for the models and observational datasets are calculated over the same region this is unlikely to affect the general conclusions. Multi-model means are calculated by first averaging all the realizations for each individual model (where appropriate), and then averaging the resulting ensemble-means. The mass budget terms are defined in Appendix E of Notz et al. (2016), and summarised here for completeness:

- Basal growth: ice growth at the base of existing ice.
- Frazil ice formation: ice formation in supercooled open water.
- Top melt: melting at the top surface of the ice.
- Basal melt: melting at the base of the ice.
- Lateral melt: melting at the sides of the ice.
- Snowice: ice formation due to the transformation of snow to sea ice due to surface flooding.
- Evapsubl: the change in ice mass due to evaporation and sublimation.
- Advection: the change in ice mass due to ice being advected into or out of the analysis domain.

The monthly mean data was calculated for the period 1960-2100 from model integrations using the CMIP6 Hist forcing scenario for the period 1960-2014, and SSP5-8.5 thereafter (Gidden et al, 2019). The SSP5-8.5 scenario

was primarily chosen because for the majority of participating modelling centres this was the first scenario to be run, but it also has the advantage of being the scenario with the highest warming signal. This means that we see relatively large changes in the ice state and the budget terms during the 21$^{st}$ century, and differences between the model budget terms are likely to be more pronounced.

In order to better understand the relative roles of atmospheric forcing and sea ice model physics, results from three

forced ocean-ice simulations are also included. All of them use the same ocean and sea ice models as the UK CMIP models (HadGEM3-GC31-LL, HadGEM3-GC31-MM and UKESM1-0-LL), with changes to both the parameter settings and the atmospheric forcing datasets. Further details of the settings and forcings used can be found in Table 3.

We also compare the modelled ice state with a number of observational and reanalysis datasets. We calculate the ice area over our analysis domain (Fig. 1) using monthly mean values for the years 1979 to 2015 from 3 observational products so that we take account of observational uncertainty:

- The second version of the global sea-ice concentration climate data record (OSI-450) from the European Organisation for the Exploitation of Meteorological Satellites (EUMETSAT) Ocean Sea Ice Satellite

Application Facility (http://osisaf.met.no/p/ice/ice_conc_cdr_v2.html, OSI-SAF, 2017; Lavergne et al., 2019).

- The first and second versions of the Met Office Hadley Centre's sea ice and sea surface temperature data set: HadISST1.2 (https://www.metoffice.gov.uk/hadobs/hadisst/, Rayner et al., 2003) and HadISST.2.2 (https://www.metoffice.gov.uk/hadobs/hadisst2/, Titchner and Rayner, 2014).

For ice mass we use monthly mean sea-ice thickness for the years 1960 to 2019 from the Pan-Arctic Ice-Ocean Modeling and Assimilation System (PIOMAS) (http://psc.apl.uw.edu/research/projects/arctic-sea-ice-volume-anomaly/data/, Zhang and Rothrock, 2003) to calculate the ice volume, which is then converted to mass using the constant value of 917 kg m$^{-3}$ for the ice density used in PIOMAS. While this is not an observational dataset, it provides a useful reference as it has been well studied and validated against observations (for example Stroeve et

al., 2014, Wang et al., 2016), and has a well-quantified measure of uncertainty (Schweiger et al., 2011).

**3. Model inter-comparison of mean sea ice state**

We first compare the ice area and mass simulated by the different models. We consider the seasonal cycle for the

period 1990-2009, in order to compare model results to present-day observations and the PIOMAS reanalysis. We also consider the evolution of the ice area and mass from 1960 to 2100, so that we can put the budget changes into context. Note that the values of ice area and mass shown here, both for the models and observational datasets, are values for the Arctic domain shown in Fig. 1a, and do not represent the whole ice-covered region.

**3.1 1990-2009 seasonal cycle**

During the period 1990-2009, there is a spread of 0.7 million km$^2$ in the March ice area simulated by the models (Fig. 1a). Despite the winter ice cover being bounded to some extent by the analysis region, the modelled values

range from 8.4 to 9.1 million km$^2$, compared with a range of 8.2 to 8.7 million km$^2$ for the three observational datasets. The observational range as quoted here is the spread in the +/- 1 standard deviation limits for the three datasets. In September we see a much larger spread in modelled ice area (3.4 million km$^2$), although most models fall within the observational range, which is larger for September than for March. One model in particular (NCAR_CESM2_CAM) has an especially large seasonal cycle and low ice area in September, possibly because of its relatively low ice mass (Fig 1b and discussed below). The magnitude of the modelled seasonal cycle in ice area varies between 3.2 and 6.0 million km$^2$. The observational datasets have their minimum ice area in September, and while 4 of the models clearly capture this, the remainder have their seasonal minimum (as calculated over our analysis region) in August or have August and September values that are very similar.

There is a large spread in modelled values of the ice mass, with the differences between models being much larger than the variability suggested by the PIOMAS reanalysis for all months (Fig. 1b). For example, in March the PIOMAS range (+/- 1 standard deviation) is 16x10$^3$ to 20x10$^3$ Gt, while the CMIP6 model values range from 13x10$^3$ to 29x10$^3$ Gt. While there are some differences in the magnitude of the seasonal cycles of ice mass, this is not as pronounced as it is for the ice area as the model spread in ice mass is relatively consistent all year. The magnitude of the seasonal cycle ranges from 8.1 x10$^3$ to 12.1x10$^3$ Gt, compared to the value of 10.1x10$^3$ Gt for the PIOMAS reanalysis. All models have their seasonal maximum ice mass in May, and minimum in September, consistent with the PIOMAS reanalysis. The models with the largest ice mass tend to be those with the smallest seasonal cycle in sea ice area, and vice versa. This is probably because a model with a smaller mass of sea ice is likely to have relatively thin ice, and so more ice cover would be lost for a given reduction in ice mass.

**3.2 Evolution from 1960 to 2100**

The models show a wide range of rates of decline in both March and September ice area (Fig. 2a). For example, the date at which the September ice area first falls below 1 million km$^2$ varies from 2019 to 2062 overall, although it is between 2025 and 2040 for the majority of models. Note that these dates are likely to be earlier than if they were calculated for the whole ice-covered region, as our domain excludes the Canadian Archipelago. By the end of the 21$^{st}$ century, all the models have completely lost their September ice cover. The evolution in March ice area starts to show a divergence between the models from about 2040, increasing significantly from about 2070. Many models show a steepening in the rate of decline of March ice area later in the 21$^{st}$ century, after their summer ice cover has melted out. Bathiany et al. (2016) found a similar steepening in the rate of winter decline for CMIP5 models. By the end of the 21$^{st}$ century there is a large spread in modelled March ice area: the fraction of ice area lost relative to 1960-89 ranges from 8% to 90%. The models with the fastest decline in September ice area do not necessarily also have the fastest decline in March ice area.

There are large differences between the models in terms of the evolution of their ice mass, illustrated here for March (Fig. 2b). We do not show the evolution of September ice mass as, in contrast to the ice area, the relative rates of decline between the models are similar for both March and September. Between 1960 and 2030, some models show an ongoing decline in ice mass, others have a period where the ice mass remains relatively stable before it starts to decline, and a few models show an increase in ice mass before the decline. By the 2030s the models have a much lower spread in values of ice mass, with a range of 8.7 x10$^3$ Gt, compared to 16.4 x10$^3$ Gt during the reference period 1960-89. By this stage there is very little mass of summer ice remaining, and so the winter ice mass is limited to the amount that can grow during a single season. Some models show a distinctive

slowing in the rate of decline of ice mass later in the 21$^{st}$ century, whereas other models show a more uniform rate of decline to the end of the 21$^{st}$ century. However, it is likely that beyond 2100 these models would also show a slowing in their rate of decline as the ice mass reduces further.

### 3.3 Ice state in a global context


To help put the ice state and its evolution in a wider context, we consider the evolution of global-mean near-surface temperature for the models (Fig. 3). We compare this model data with annual mean temperature for the years 1960 to 2019 calculated from the HadCRUT4.6.0.0 observational dataset (https://www.metoffice.gov.uk/hadobs/hadcrut4/, Morice et al., 2012). The majority of the models considered

here are warming more quickly than the observations suggest. The mean warming for 2014-19 relative to 1960-89 ranges from 0.6° to 1.4° for the models, compared to the observed 95% confidence range of 0.6° to 0.8°. By the end of the 21$^{st}$ century, the warming relative to the 1960-89 mean ranges from 3.9° to 7.2°. Models with a larger decrease in March ice area by the end of the 21$^{st}$ century tend to be those with a larger increase in global mean temperature.

A recent study of 40 CMIP6 models (SIMIP Community, 2020) found that the majority (29 out of 40) simulate too small a reduction in sea ice area per degree of warming, so their sea ice sensitivity is too low. As a result, very few of the models simulate both a plausible rate of sea ice loss and a plausible rate of global warming. Of the CMIP6 models involved our study, the following models were reported by SIMIP Community (2020) as capturing both: ACCESS-CM2, GFDL-ESM4, MRI-ESM-2.0 and NorESM2-MM, although it is worth noting that this

calculation used only the first ensemble member for each model, whereas in our study we use several ensemble members for some of the models (Table 1). This subset of models are amongst those with the smallest temperature increase by the end of the 21st century in Fig. 3.

### 4. Mean sea ice mass budget for 1960-1989


We now consider the mass budget of the Arctic sea ice, as defined in Sect. 2 and also in Notz et al. (2016). We start by looking at the model data for a reference period 1960-89, chosen as a time when the ice cover and mass is relatively stable. Note that this is not the same time period used for the ice area and mass in Fig. 1, which was chosen to cover a period with more observational data. We first consider the multi-model mean budget, and then

look at the differences between models.

### 4.1 Multi-model mean

Figure 4a shows the budget terms for the mass budget of the Arctic sea ice, averaged over the analysis region

shown in Fig. 1a, for the multi-model mean for the reference period 1960-89. The black line shows the total amount of ice growth or loss each month, with net ice loss occurring from May to September, and net ice growth from October to April. Most of the increase in ice mass results from growth at the base of existing ice, which occurs between September and May and represents 83% of the total annual ice growth (Fig. 4b). Frazil ice formation in open water accounts for 16%, with the small remainder due to snow-ice formation. Most of the ice

loss occurs during the summer, with 52% of the annual mean ice loss caused by basal melting due to heat from the ocean, and 25% by melt at the ice surface (Fig. 4b). The monthly maximum ice melt occurs in July, with both top and basal melt peaking at this time, and the basal melt continues further into the late summer that the top melt

(Fig. 4a). Ice lost by advection out of the region accounts for 19% of the total (Fig. 4b), and this is likely to be dominated by loss through the Fram Strait. The advective ice loss occurs all year, and although it is greater during the winter than the summer, the magnitude of the seasonal cycle is far smaller than that for either the top or basal melting (Fig. 4a).

In summary, for the 1960-89 multi-model mean the dominant process causing ice growth is basal ice growth, followed by frazil ice formation, and the dominant processes causing ice loss are basal melting, top melting and advection out of our analysis region. For the remainder of the paper we focus on these five main budget terms, as the remaining smaller terms do not contribute significantly to the mass budget for any of the models considered.

**4.2 Inter-comparison of the CMIP6 models**

Figures 5 and 6 show the annual means and seasonal cycles of the main budget terms for each individual model. Note that some models do not generate all the terms, and some models have terms missing (see Table 1). It is striking to see how similar the model budgets are, at least in a broad sense (Fig. 5). For example, in all the models we see:

- more basal ice growth than frazil ice formation (Fig. 5),
- virtually no ice growth between June and August (Figs. 6a and 6b),
- more basal melting than top melting (although the difference is small for the UK models and CSIRO_ACCESS-CM2) (Fig. 5),
- a maximum in ice melting in July, with a peak in both top melting and basal melting (Figs. 6c and 6d),
- basal melt continuing later into the autumn than top melting (Figs. 6c and 6d),
- and a relatively symmetric seasonal cycle of total ice growth and melt, centred around the maximum net ice loss in July (not shown).

However, there are also some notable differences between the model budgets, which we describe below.

Ice growth

The ratio between basal ice growth and frazil ice formation varies significantly between the models. For example, for both the GFDL models almost all the annual ice growth is due to basal ice growth, whereas for the EC-Earth3 models 73% of the annual growth is due to basal growth, and 25% to frazil ice formation. This different partitioning appears to relate to specific settings within the sea ice models, in particular the minimum thickness of frazil ice that is allowed to form. The lower the value of the minimum frazil ice thickness, the more quickly the frazil ice growth can transition to basal growth. This is discussed further in Section 5.

If we consider the total amount of winter ice growth (here taken as the sum of the frazil and basal growth terms), the spread in modelled values is $3.9 \times 10^3$ Gt, compared to the larger range of $5.9 \times 10^3$ Gt for the basal growth alone. The month in which the amount of frazil ice formation peaks varies from October (UK models and CSIRO_ACCESS-CM2) through November (EC-Earth3, EC-Earth3-Veg and CM6A-LR) to December for the remaining models (Fig. 6b). The amount of basal growth peaks in December for most of the models (Fig. 6a).

Ice melt

The relative amount of basal and top melting also varies considerably between the models, with top melting ranging from 28% to 91% of the amount of basal melt. The UK models (HadGEM3-GC31-LL, HadGEM3-GC31-MM and UKESM1-0-LL) together with CSIRO-ACCESS-CM2 have almost as much ice lost by top melting as by basal melt, in contrast to the other models which have considerably more basal melt than top melt (Fig. 5). The models agree very well on the amount of top melting late in the season (Aug-Sep). However, the onset of melt (May-June), and peak in melting in July shows much more variability (Fig. 6c). This is consistent with the former being mainly related to the solar zenith angle, whereas the latter is also related to other factors like ice/snow surface temperature and surface albedo. Seven out of the 15 models have more top melting than basal melting in July, and these are the models with most top melting overall. Some models have a fairly symmetrical peak in basal melting (for example the EC-Earth3 models and also the NCC_NorESM2-LM model.), whereas some have a more pronounced 'tail' of melt further into autumn (for example MRI_ESM2.0, NCAR_CESM2_WACCM) (Fig. 6d).

Ice advection

All the models for which we have the dynamics term show a net advection of ice out of the analysis region (Fig. 5). This is dominated by export through the Fram Strait. The net ice loss by advection varies from $1.2 \times 10^3$ to $3.7 \times 10^3$ Gt per year between the models and comprises between 9% and 30% of the total annual ice loss. There is no strong agreement between models as to how this export varies during the year (Fig 6e). Five of these 13 models have a minimum advective ice loss in August, but overall there is considerable variability between the models in terms of the amount and timing of the annual ice export.

**5. Understanding differences between the CMIP6 models**

Having described the similarities and differences in CMIP6 model ice budgets for the reference period 1960-89, we now explore the extent to which these budget differences can be related to differences in model formulation and ice state. The sea ice mass budget in climate models is influenced by both the sea ice physics and also the interaction with the atmosphere and the ocean, meaning it can be difficult to isolate the impact of, say, a particular parametrisation when inter-comparing different models. Hence to aid our understanding we also draw on information from a set of ocean-ice experiments, where the atmospheric forcing and sea ice physics are varied independently. In this section, we also try to identify the influence of the sea ice model physics on the different mass budget terms.

**5.1 Description of the forced ocean-ice experiments**

We consider three integrations using a forced ocean-ice model with the same sea ice and ocean components as the HadGEM3-GC31-LL model. The advantage of using a forced model is that we can look at the impact of changing sea ice physics or atmospheric forcing independently (Table 3). The main disadvantage is that it does not represent atmosphere-sea ice and atmosphere-ocean feedbacks. It is noteworthy that the default forced simulation differs strongly from the HadGEM3-GC31-LL simulation despite the same sea ice settings (Fig. 7). In

contrast to HadGEM3-GC31-LL, sea ice mass and area are smaller in the forced simulation than in PIOMAS, and HadISST and OSI-SAF respectively. The stronger top and basal melt in the forced simulation (Figs. 8c and 8d) leads to a stronger annual cycle and smaller sea ice area during summer (Fig. 7a). This suggests that the interaction with the atmosphere is responsible for the overestimation of sea ice mass in HadGEM3-GC31-LL.

Applying changes to the sea ice physics which results in improved sea ice evolution in a stand-alone sea ice
simulation (Schroeder et al., 2019), reduces the basal melt (Fig. 8d) and slightly increases the basal growth (Fig. 8a), which in turn increases sea ice area and mass (Fig. 7). The new settings include the elastic anisotropic plastic, rheology (Tsamados et al., 2013), the bubbly conductivity formulation from Pringle et al. (2007), a simple scheme to account for the loss of drifting snow, increased longwave emissivity of sea ice from 0.95 to 0.976 and the maximum meltwater added to melt ponds, rfracmax, is reduced from 100% to 50% (Schroeder et al., 2019). These
changes do not bracket the whole range of physics seen in the CMIP6 models, but the forced experiments do include physics not used in any of the CMIP6 models included in this paper (the EAP rheology and bubbly conductivity). The differences regarding the sea ice area and mass, and the basal melting, are striking given we apply the same sea ice model and the changes do not bracket the full potential variations in sea ice physics. A similar effect was found by Massonnet et al. (2018), who showed that using the simplest Semtner 0-layer model
and varying only one parameter (ice albedo) could explain a great fraction of CMIP5 model spread.

Replacing the CORE forcing which is based on NCEP reanalysis (Large and Yeager, 2009), with DFS forcing based on ERA-interim reanalysis (Dussin et al, 2016) increases the top melt (Fig. 8c) and decreases the basal melt (Fig. 8a), resulting in higher sea ice area and mass. Interestingly, differences between the CORE and DFS forced experiments in late summer suggest that the winds and large-scale atmospheric circulation are having a larger
impact than sea ice physics on sea ice dynamics (Fig. 8e). It is likely that there is more diversity in the atmospheric forcing in the CMIP6 models than in these two forcing datasets, which are both based on re-analyses. The CMIP6 models also represent the two-way atmosphere-ice interactions, which the forced models do not. Looking at the different components of the mass budget, these three forced experiments can nearly explain the full CMIP6 model spread for basal sea ice melt, although only a small part of the differences in top melting and sea ice growth. These
experiments confirm that both the sea ice physics and the atmospheric forcing are key factors in determining what the ice state and mass budget look like, and can produce changes of comparable magnitude.

### 5.2 Individual budget terms in the CMIP6 models

We now consider each of the mass budget terms in turn, to see to what extent we can find factors that might explain the differences between the CMIP6 models. When looking at differences in model formulation, we have focussed on the physics and parameter choices made within each sea ice component. The sea ice models used here share a number of key parametrizations, and often provide a choice of different schemes to choose from. For
example, this study includes models using CICE, SIS and LIM sea ice components that all use the BL99 (Bitz and Lipscomb, 1999) thermodynamic scheme, while there is another that uses CICE with the T13 (Turner et al, 2013) thermodynamics. The sea ice models used in the CMIP6 models considered here can be grouped according to a

number of key model parametrizations and settings (Table 4), and for each budget component, we consider its relationship with these (Fig 9), as well as the ice state (March and September ice area, and annual mean ice mass) and atmospheric near-surface temperature (not shown). Note that the mean values of ice state and atmospheric temperature discussed here are calculated for the reference period 1960-89, and so (for ice state) do not exactly match the data as shown in Fig. 1.

Basal Growth

The two factors most influencing the amount of basal growth in the models considered here are the thermodynamic scheme and the melt-pond formulation. The models using CICEb, LIM3 and SIS2 use the BL99 thermodynamic scheme, and 7 out of 9 of these have more basal growth than the 4 CICEa models using T13. This is consistent with sensitivity experiments using the CESM2_CAM model (Bailey et al, 2020) (B20), which found more basal growth with the BL99 scheme than T13. The 8 models that use CICE all have prognostic melt-ponds. Those using CICE 5.1.2 GSI8.1 (CICEb) have the F12 scheme (Flocco et al, 2012), and have more basal growth than those using CICE5.1.2 (CICEa) with the H13 melt pond scheme (Hunke et al, 2013) (Fig.9a). The models with parametrized melt ponds show a wide range of values of basal growth. We have not found a relationship between the amount of basal growth and the ice state or global atmospheric temperature

Frazil ice formation

The main factor affecting the amount of frazil ice formed by each model is the value chosen for the minimum thickness of frazil ice. The GFDL-CM4 and GFDL-ESM4 models with the SIS2 sea ice component have no minimum thickness, which means these models can grow arbitrarily thin frazil ice. This means the frazil ice can quickly transition to congelation growth, and these models form less than $0.04 \times 10^3$ Gt/year of frazil ice. Of the remaining models, those that can form thicker frazil ice tend to form more frazil ice and correspondingly less ice via basal growth. There are 4 models using LIM3 with a minimum frazil ice thickness of 10cm, and these form between 2.5 and $3.5 \times 10^3$ Gt/year of frazil ice (Fig 9b). The remaining models have a minimum frazil ice thickness of 5cm, and 6 out of 8 of them form less ice than the models with a 10cm minimum thickness. While it is not unexpected that the minimum frazil ice thickness affects the amount formed within a particular model, it is notable that this relationship is seen so strongly across the majority of models considered here, despite all the other differences in model formulation. The sea ice component affects the month in which the frazil ice formation peaks (Fig. 6b): models using CICEb have a maximum in October, those using LIM3 in November, and those using CICEa in December.

The B20 study found that the T13 thermodynamic scheme tends to produce more frazil ice than BL99, however this difference is not evident in the models considered here. Another finding from B20 is that model configurations with larger extents tend to have more frazil formation, as the biggest differences are seen in the marginal ice area. Here, while the 4 models with the largest March ice area do form relatively large amounts of frazil ice, there is not a strong correlation when all the models are considered, likely due to the areal domain considered in this study.

Top melt

We have found links between the amount of top melting and the melt-pond formulation, the thermodynamic scheme, the treatment of incoming shortwave radiation and the global mean near-surface temperature. Eight of the 15 CMIP6 models have a relatively small amount of top melting (Fig. 9c), melting between 2.1 and 2.5 $\times 10^3$ Gt/year of ice. The remaining 7 models melt between 3.5 and 4.9 $\times 10^3$ Gt/year. Of the models with prognostic melt-pond schemes, those models using H13 (CICEa) have less top melting than those using F12 (CICEb). For models with a parametrized representation of melt ponds there is no clear separation in the amount of top melt. A unique feature of the models using CICEb compared to the other CMIP6 models considered here is that they do not allow any of the incoming shortwave radiation to penetrate through the sea ice to the ocean. Development experiments using an updated version of the UK model show that there is a large reduction in top melt (around 20%) when the penetration of shortwave radiation is included (Blockley, private communication), consistent with the relatively high amount of top melting seen here in the models without this feature.

The 4 models using CICEa with the T13 thermodynamic scheme have a relatively small amount of top melt. The B20 study found more top melting with the BL99 scheme than with the T13 scheme. Here we find that a majority of the models using BL99 (6 out of 9) have more top melt than the models using T13, with the remaining 3 having a similar amount of melt to the T13 models. Six out of the 7 models with the larger amount of top melting have relatively low March ice area: there is no obvious relationship with September ice area or annual ice mass. The majority of the models with less top melting (7 out of the 8) have a relatively high global near-surface temperature.

Basal melt

Models with a lower September ice area tend to have more annual basal melt. This would be consistent with those models having a greater potential for atmospheric heating of the upper ocean, due to the lower ice cover. Models with a smaller annual mass of ice also tend to have more basal melt. These findings are consistent with the results from the forced ocean-ice experiments. The amount of basal melt will also be affected by the ocean temperature and heat budget of the upper ocean, but an analysis of these factors is outside the scope of this study. The models with parametrised melt ponds tend to have more basal melt than those using CICE with prognostic melt ponds (Fig. 9d). For those models using prognostic schemes, models using H13 (CICEa) tend to have more basal melt than those using F12 (CICEb), although it is not a very marked difference. The two models using the zero layer and MK89 thermodynamic schemes (LIM2 and COM4.4) have more basal melt than the other models. The B20 study found more basal melt with the BL99 thermodynamic scheme than with T13, but this is not evident in the CMIP6 models. Models using the BL99 scheme (CICEb, LIM3 and SIS2) have a relatively wide range of basal melt (from 4.5 to 7.58 $\times 10^3$ Gt/year) compared to those using CICEa with T13 (6.1 to 6.9 $\times 10^3$ Gt/year), but there is no clear separation in the amount of basal melt between models using these two thermodynamic schemes.

In section 4.2 it was noted that the majority of models have notably more basal melt than top melt (Fig. 5). The exceptions are the UK models and CSIRO_ACCESS-CM2, where the difference is much smaller. This is likely to be due to the lack of penetrating solar radiation mentioned above, although these models do also share the same sea ice (CICEb) and atmosphere (MetUM) components, so there are potentially other factors which could also be responsible.

Ice advection

Models with a larger annual mean mass of ice tend to lose more ice each year by advection out of the analysis region. There is also a similar relationship with September ice area, and to a lesser extent with March ice area. This is perhaps not surprising – if there is more ice in the Arctic basin there is more available to be transported out. There is some evidence that the atmospheric component plays a role: for example, the models using MetUM and NorCAM6 atmospheres have the most advective ice loss (Fig. 9e). Apart from that we have found no clear link between the ice loss by advection and the model formulation. This is consistent with the findings from the forced experiments, where winds and large-scale atmospheric circulation have a larger impact than sea ice physics on sea ice dynamics.

Summary

We have identified a number of potential links between model physics and ice state, and the major components of the ice mass budget amongst the CMIP6 models for the reference period 1960-89. For models with prognostic melt-ponds, the choice of scheme may affect basal growth, basal melt and top melt. The F12 scheme tends to be associated with more basal growth and top melt but less basal melt than H13. One notable result is that the amount of frazil ice formation is strongly dependant on the value chosen for the minimum frazil ice thickness, despite all the other differences in model formulation.

The thermodynamic scheme used is related to the amount of basal growth and top melt, with models using the BL99 scheme having more basal growth and top melt than those using T13. This result is consistent with Bailey et al (2020), who studied the impact of the BL99 and T13 thermodynamic schemes within a single climate model. This study also found that the T13 thermodynamic scheme results in more frazil ice formation and basal melt than BL99. We cannot detect this amongst the CMIP6 models, possibly because of the many other differences in model formulation.

Models using the MetUM and NorCAM6 atmospheric component tend to have more dynamic ice loss than the other models, which is potentially consistent with the finding from the forced experiments that winds and large-scale atmospheric circulation are important in determining the amount of dynamic ice loss. These models also have higher ice mass than most of the other models, so for the same sea ice drift they will export more sea ice mass from the domain. We have not found any other clear links between the mass budget components and the atmospheric model used. This is perhaps surprising given the results from the forced experiments. However, it may be that the internal atmospheric variability masks this within the CMIP6 model experiments. It is also worth noting that all the models using the F12 melt-pond scheme have a common atmospheric model (MetUM), so this may also be a factor affecting the top melt, basal melt and basal growth.

In summary, we have found a number of relationships between model formulation and components of the ice budget that hold for all or most of the CMIP6 models considered here.

## 6. Projections of the sea ice mass budget during the 21st century

We now consider how the mass budget of the CMIP6 models changes during the 21$^{st}$ century as the ice cover and mass declines, and the environment warms. We first look at the evolution of the budget terms for the multi-model mean, and then consider the differences between individual models.


**6.1 Multi-model mean**

The magnitude of each budget term tends to decrease with time, consistent with the reducing mass of the ice (Fig. 10a). Throughout the time period considered here there is a greater mass of ice formed by basal growth than by

frazil ice formation, and more basal than top melt. The amount of ice lost by advection tends to decrease relatively quickly compared to the decline in ice growth and melt. Some of the terms initially increase before they start to decline, and this is seen more easily in Fig. 10b, where each term is plotted as an anomaly relative to the 1960-89 mean. We will now consider each of the main budget components in turn.

Basal ice growth

The multi-model mean initially shows a very small increase in the amount of basal growth, followed by an ongoing decrease throughout the 21$^{st}$ century (Fig 10b). From the 2030s onwards there is less basal growth than during the reference period, and by the 2090s the amount of basal growth has reduced by 67% relative to the reference period. The decadal means shown in Fig. 10 are a combination of opposing changes occurring at different times of year,

illustrated in Fig. 11 for the decade 2040-49. During the winter there is an increase in the amount of basal growth compared to the reference period (Fig. 11b), consistent with the ability of thinner ice to grow more quickly than thicker ice (Bitz and Roe, 2004). This is offset by a decrease in the amount of basal growth in the autumn, due to there being a smaller area of ice (relative to the reference period) over which basal ice can grow. Warmer ocean temperatures compared to the reference period also reduce both frazil and basal growth. Initially the winter

changes dominate, but from the 2030s onwards the reduction in autumn basal growth dominates. In Fig. 10c we remove the direct impact of the declining ice area on the basal growth by plotting the mean growth per unit area of the ice. As the climate warms there is an increasing amount of basal ice growth per unit area of ice, consistent with the ability of thinner ice to grow more quickly than thicker ice. Later, as the warming continues, the basal growth per unit area of ice starts to decrease as the rising temperatures act to suppress the ice growth.


Frazil ice formation

As the climate warms there is a small increase in the total amount of frazil ice formation relative to the 1960-89 reference period, then from the 2050s onwards there is a reduction (Fig. 10b). By the 2090s the amount of frazil ice formation has reduced by 58%. Again, these changes result from opposing changes at different times of years.

During the winter there is more frazil ice formation as the climate warms, probably because as the ice area declines there is a greater area of open water in which frazil ice can form. This is offset by a reduction during the autumn, consistent with the ocean warming so that an increasing area of the ocean either no longer falls below the freezing temperature or does so for a shorter period of the year. As the warming continues, the reduction in frazil ice formation in the autumn becomes more dominant and continues later into the year, while the extra frazil ice

formation during the winter ceases (not shown). Again, these changes are consistent with the ongoing warming of the ocean.

Top melt

For the multi-model mean there is an increase in the total amount of ice melted at the top surface relative to the 1960-89 reference period, which continues until the 2040s (Fig. 10b). During the reference period, there is a symmetrical seasonal pattern of top melt, with a peak in July (Fig. 11a, faint blue line). As the climate warms, there is more top melt earlier in the melt season, and the peak shifts to June. There is less top melt during July and August due to the larger reductions in ice area at this time of year, leaving a smaller area over which ice melt can occur. By the 2060s, as the ice area continues to decline, the reductions in melt during July and August start to dominate so that the total amount of top melting is less than during the reference period. (Fig. 11b). By the 2090s the amount of top melt has reduced by 68% relative to the reference period. The amount of top melt per unit area of the ice initially increases as the climate warms (Fig.10c), consistent with the warming atmosphere. However, towards the end of the 21$^{st}$ century, the top melt per unit area starts to decrease again, presumably because by this stage the ice cover that remains is restricted to relatively high latitudes, and so receives less solar radiation.

Basal melt

The total amount of basal melt evolves in a similar way to the top melt (Fig. 10b), with an initial increase relative to 1960-89 persisting until the 2040s, followed by a reduction relative to 1960-89 from the 2060s. Again, there are different changes occurring at different times of year (Fig, 11). During the reference period the peak in basal melt occurs in July. As the climate warms, the seasonal cycle becomes less symmetric as the amount of basal melt earlier in the melt season increases (consistent with ocean warming), while later in the year it decreases relative to the reference period due to their being a smaller area over which the melt can occur. By the 2060s the peak in basal melt occurs in June rather than July (not shown). By the 2090s the basal melt has decreased by 54% relative to the reference period. The amount of basal melt per unit area of the ice (Fig. 10c) continues to increase throughout the 21$^{st}$ century, consistent with an ongoing warming of the upper ocean.

Ice advection

The amount of ice lost by advection declines relatively quickly compared to the other terms (Fig. 10a) and has reduced to virtually zero by the 2090s. Compared to the other main budget terms, the changes do not show a distinct seasonal pattern. Further insight on the changes in ice advection would require an analysis of winds and large-scale atmospheric circulation, and the associated ice motion, which is outside the scope of this study.

Summary of multi-model mean changes

Initial reductions in the ice mass are due to extra top and basal melting, partially offset by a reduction in ice lost by advection out of the Arctic basin. Later in the 21$^{st}$ century the declining ice area has a significant impact on the size of the budget terms. For example, the amount of basal melt per unit area of the ice continues to increase throughout the 21$^{st}$ century (Fig. 10c), but the total amount of ice lost by basal melt reaches a maximum in the 2020s and declines after that because there is a reduced area of ice over which the melting can occur. By the 2070s the ongoing ice loss is primarily due to reductions in winter basal ice growth, with some reduction in frazil ice formation (Fig, 10b).

There is a distinct seasonal pattern to the budget changes. During the 2040s, the budget changes leading to extra ice loss occur primarily during May and June, when there is extra top and basal melt, and October and November when there is reduced basal growth (Fig. 11b). These changes are partially offset by reduced top and basal melt during August and increased basal growth during January-March. This seasonal pattern of change broadly persists through the 21$^{st}$ century, although by the 2070s there is also reduced top and basal melt in July, and the extra winter basal ice growth has ceased (not shown).

**6.2 Differences between the CMIP6 models**

We now consider how the budget terms for the individual models evolve, and the similarities and differences between the models. First, we consider each term individually, and then we look at the overall budget.

Basal growth

The majority of models show a gradual decrease in the total amount of basal ice growth as the climate warms (Fig. 12a). For some of the models there is initially an increase before the decline, and this is most pronounced for models with the slowest decline in ice cover, particularly during winter. For example, the NorESM2-MM model has one of the slowest rates of decline of ice area of the models considered here (Fig. 1a), and the amount of basal growth remains higher than the 1960-89 reference value until the 2080s. In contrast, the IPSL_CM6A-LR model has one of the fast rates of decline of ice cover, and only a marginal increase in basal growth relative to the reference period before declining from the 2020s. Both these models show an increase in winter basal ice growth by the 2010s, as the ice thins (not shown). For IPSL_CM6A-LR, this is offset by reduced growth in autumn due the smaller ice area, whereas for NorESM2-MM the reduced autumn ice growth is not evident until the 2060s (not shown). Overall, models with the largest decline in basal ice growth by the end of the 21$^{st}$ century tend to be those with the larger decline in winter ice cover (Figs. 2a and 12a).

Frazil ice formation

As the climate warms, almost all the models show the initial increase in the total amount of frazil ice formation following by an ongoing decline as seen in the multi-model mean (Fig. 12b). The exception is the IPSL_CM6A-LR model, which has no initial increase in frazil ice growth relative to the reference period, and the GFDL models which have a very tiny amount of frazil ice formation that hardly changes throughout the 21$^{st}$ century. The timing of the changes in frazil ice formation varies considerably between the models. For example, the decade in which the amount of frazil ice formation becomes less than the reference value varies from the 2000s for the IPSL_CM6A-LR model to the 2070s for CSIRO_ACCESS-CM2.

Top melt

All the models show an initial increase in the total amount of ice melted at the top surface relative to 1960-89, which continues until at least the 2040s for the majority of models. (Fig. 12c). As the warming continues and the ice cover reduces, all the models show a decline in the amount of top melt. By the end of the 21$^{st}$ century, all the models except NorESM2-MM have less top melting than they did during the 1960-89 reference period. The timing of the maximum amount of top melt varies between the models (Fig. 12c) and is related to the rate of decline of

the ice area. For example, the NorESM2-MM model has a relatively slow decline in ice cover. Its top melting continues to increase until the 2050s, and by the end of the 21$^{st}$ century it still has more ice lost due to top melting than during the reference period. In contrast, the IPSL_CM6A-LR model has a relatively rapid decline in ice cover, and a maximum top melt during the 2000s, which is the earliest of all the models. The models with the largest decline in top melt by the end of the 21$^{st}$ century tend to be those with the largest reductions in ice area.

Basal melt

For almost all the models, the total amount of basal melting starts to increase as the climate warms (Fig. 12d), then decreases later in the integration as the ice cover shrinks. The timing of the peak in basal melt ranges from the 2010s to the 2060s, and by the end of the 21$^{st}$ century almost all the models lose less ice each year by basal melting than they did during the reference period, and the models with the greatest reduction in basal melt tend to be those with the largest reduction in ice area. In common with the multi-model mean, all the models show an ongoing increase in the amount of basal melt per unit area of the ice throughout the 21$^{st}$ century (not shown).

Ice advection

For all the models, the amount of ice lost by advection declines during the 21$^{st}$ century (Fig. 12e), and this decline occurs relatively quickly compared to the other terms. The models with the largest reductions in advective ice loss by the end of the 21$^{st}$ century tend to be those with the largest reduction in ice mass.

Summary of differences between the CMIP6 models

Having looked at each of the main budget components in turn, we have found that for each model the budget components evolve in a broadly similar way to the multi-model mean during the 21st century. The primary difference between the models is the timing and magnitude of the changes, many of which are strongly related to the rate at which the ice cover reduces.

Figure 13 shows the changes in the main budget terms for each model in turn, so that we can see how the terms change in relation to each other. There are some key features seen in both the multi-model mean (Fig. 12b) and many of the individual models:

- The amount of ice advection changes relatively quickly compared to the other main budget terms
- Models with a more rapid decline in basal ice growth tend to also have a more rapid decline in top and basal melt.
- Initially the ice loss is due to extra top and basal melting relative to the reference period, which is partially offset by reduced advective ice loss, and an increase in basal ice growth.
- Towards the end of the 21$^{st}$ century, and after the Arctic Basin becomes virtually ice free at the end of the summer, the ice loss is primarily due to reduced basal ice growth, plus reductions in frazil ice growth, partially offset by reductions in top and basal melt and advective ice loss.

The latter changes are more pronounced in the models with a faster decline in ice cover. The NorESM2-MM model has the slowest decline in ice area, and by the end of the 21$^{st}$ century this budget shows a rather different combination of changes. The ice loss at this stage is due to reductions in basal growth and frazil ice formation, together with extra basal melting. These changes are partially offset by the reduced advective ice loss, with little change at this stage in the amount of top melt relative to the reference period. However, this combination of

changes is seen in other models earlier in the 21st century, so it is quite possible that this budget would evolve in a similar way to the other models as the warming continues into the 22nd century.

If we now plot the changes in the budget terms as a function of the ice state rather than time (Fig. 14), we highlight the similarities in the evolution of the model budgets rather than their differences, and we can identify robust relationships common to all the models. The change in basal ice growth shows a very consistent relationship with the ice mass for all the models, with the decrease relative to the reference period starting once there is approximately $5 \times 10^{-3}$ Gt of ice remaining (Fig. 14a). The changes in basal growth, and top and basal melt are all related to the ice area, with a maximum amount of basal and top melt occurring when the annual mean ice area of approximately 6 million $km^2$ (Figs 14a, b and c). The change in the advective ice loss is related to the change in ice mass (Fig. 14e).

In summary, the budget components of the models considered here evolve in a broadly similar way to each other as the climate warms, although with considerable differences in the timing and magnitude of their changes. The differences seen are consistent with differences in the evolution of the ice state, especially the ice area.

## 7. Discussion and conclusions

We have compared the mass budget of the Arctic sea ice for 15 CMIP6 models, using new diagnostics not available for previous model inter-comparison studies (Notz et al., 2016). In common with CMIP3 and CMIP5 models, the CMIP6 models we have analysed here show a large spread in their ice area and mass, both during a 'present day' evaluation period, and also as the ice declines during the 21st century (SIMIP community, 2020).

We have found broad agreement between the mean mass budget of the individual models, both in terms of the dominant processes, and also the time of year when each process is important. For the multi-model mean, the dominant processes causing annual ice growth are basal ice growth (~80%) and frazil ice formation (~20%), which both occur during the winter. The dominant processes by which ice is lost are basal melting (~50%), surface melting (~25%) and advection of ice out of the Arctic (~20%). The two first processes occur in summer, while the latter process is present all year long.

Some of the differences between the individual CMIP6 model budgets are potentially attributable to particular physics schemes or parameter settings within the sea ice model component, in particular the melt-pond formulation, the thermodynamic scheme, and the minimum thickness at which frazil ice can form. That is not to say that other sea ice physics schemes and settings have no impact on the mass budget, only that a link could not be identified using the CMIP6 models considered here.

For models with prognostic melt ponds, the Flocco et al (2012) scheme tends to be associated with more basal growth and top melt, but less basal melt than the Hunke et al. (2013) scheme. Models using the Bitz and Lipscomb (1999) thermodynamics scheme tend to have more basal growth and top melt than those using the Turner et al. (2013) scheme. The latter findings are consistent with sensitivity experiments using a single CMIP6 model (Bailey et al, 2020). The amount of frazil ice formation is related to the minimum thickness at which frazil ice can form; models that can form thicker frazil ice tend to have a greater proportion of their ice growth defined as frazil ice and correspondingly less ice mass produced via basal growth.

The atmospheric forcing also has an impact on the ice budget. In a set of forced ocean-ice experiments using the same sea ice and ocean components as one of the CMIP6 models, the winds and large-scale atmospheric circulation are important in determining the amount of dynamic ice loss, and there is some evidence of this within the CMIP6 models as well.

While it is expected that changing model physics or forcing alone leads to changes in the sea ice mass budget (as clearly demonstrated in the forced ocean-ice experiments), it is notable that we have found a number of relationships between model formulation and components of the mass budget that hold of all or most of the CMIP6 models considered here, where effectively a number of factors are varied simultaneously.

We have also looked at how the mass budget changes as the climate warms during the 21st Century. For the multi-model mean, the timing of the top and basal melting shifts, with more melt occurring earlier in the year (especially during June), and less melt later in the summer (especially during August). The amount of basal growth in the autumn reduces, but there is increased basal growth later in the winter associated with the thinner ice in the future projections compared to the present. Overall, extra ice loss in May-June and reduced growth in October-November is partially offset by reduced ice melt in August and increased ice growth in January-February.

Comparing the responses of the individual models during the 21st century, the timing and magnitude of change in the mass budget components varies considerably. However, when these components are considered as a function of the changing ice state itself, common behaviours emerge, which suggests that the mass budgets of the models are fundamentally responding in a broadly consistent way to the warming climate.

These results are broadly consistent with a previous study that considered the volume budget of the Arctic sea ice for a single CMIP5 model (HadGEM2-ES) under a range of climate forcing scenarios (Keen and Blockley, 2018). HadGEM2-ES has a similar balance of budget terms to the CMIP6 models considered here, and the budget terms evolve in a similar way as the climate warms. The choice of forcing scenario affects the timing and magnitude of the changes in the budget components, but common behaviours emerge when the changes are plotted as a function of the ice state. This work goes beyond the Keen and Blockley (2018) study by including 15 CMIP6 models rather than considering a single model.

Perhaps the most striking result of our study is the similarity in the mass budgets of the models considered here, both in terms of their mean state, and how they evolve in a warming climate. This is despite a wide spread in the values of ice area and mass simulated by these models. From this study, it is difficult to say whether the similarity in the model budgets is a good thing: does this similarity give us confidence that the models are doing the right thing, or is it an indication of how little model diversity we have?

Tables 2 and 4 demonstrate the lack of sea ice model diversity in these CMIP6 models, and it could be argued that we are not sampling enough uncertainty in the sea ice physics. For example, all the (CMIP6) models considered here use the same Elastic-Viscous-Plastic ice rheology, despite the availability of alternatives such as the Elastic-

Anisotropic-Plastic (Wilchinsky et al., 2006) or Elasto-Brittle (Dansereau et al., 2016) schemes. Including a wider range of models would help to better understand the impact of model diversity on the mass budget.

Ideally, we would be able to compare both the ice state and the budget components against observational datasets, to assess whether the models are able to generate the correct ice state for the correct reasons. While some validation of the underlying processes that determine the ice area is possible (for example Holmes et al, 2019), this remains more difficult for the mass budget terms. Recently, West et al. (2020) have processed data from mass balance buoys to provide observational estimates of vertical energy fluxes over two regions of the Arctic, and this data

should allow at least a limited validation of aspects of the model budgets in the future.

In summary, new diagnostics available for CMIP6 models have allowed a more detailed inter-model comparison of the Arctic sea ice mass budget than has previously been possible. This study has provided a first comparison of these diagnostics for a subset of the CMIP6 models by comparing budget terms integrated over the Arctic

Ocean. The model budgets are strikingly similar, but it is not clear to what extent this reflects a lack of model diversity. As more CMIP6 model data becomes available, it would be good to examine the budget of a wider range of models. Further work could also include a comparison of the mass budget of snow on sea ice, an investigation into the spatial distribution of the ice and snow budget terms, and a linking of the budget changes to changes in the wider climate, both in the Arctic and beyond.


**Appendix A: Model descriptions**

Here we provide a brief summary for the CMIP6 models used in this study. Further details can be found in the cited references.


AEMET_EC-Earth3 and SMHI_EC-Earth3-Veg

EC-Earth is an Earth System Model that comprises coupled component models for atmosphere, ocean, sea ice and land (Hazeleger et al., 2012). Both versions used in this study have the same model configuration for atmosphere, ocean and sea ice. Besides, EC-Earth3-Veg has also an interactive vegetation module. The atmosphere component

is represented by the Integrated Forecasting System (IFS), cycle 36r4, of the European Centre for Medium-Range Weather Forecasts (ECMWF).

The EC-Earth's ocean component is represented by the version 3.6 of NEMO (Madec, 2016) using the ORCA1L75 grid with a horizontal resolution of about 1 degree and 75 vertical levels. The ocean component includes the version 3 of the Louvain-la-Neuve sea ice model (LIM3; Rousset et al., 2015). LIM3 is a dynamic-

thermodynamic model with a prognostic sea ice thickness scheme (Thorndike et al., 1975; Lipscomb, 2001) defined by 5 thickness categories. LIM3 applies the elastic-viscous-plastic (EVP) rheology for ice dynamics and also uses the ORCA1L75 grid. The sea ice model thermodynamics is characterized by an energy-conserving halo-thermodynamic scheme (Bitz and Lipscomb, 1999) with 2 ice layers and 1 snow layer. The surface albedo depends on the ice surface temperature, ice thickness, snow depth, and cloudiness. The radiation absorbed by the ice

follows Beer's law and does not infiltrate into the snow. LIM3 does not account for the lateral melting. Melt-ponds are not included in the sea ice model.

The H-TESSEL scheme, cycle 36r4 (same as IFS), is used for the land surface (van den Hurk et al., 2000. In addition, EC-Earth3-Veg uses the version 4 of the Lund-Potsdam-Jena General Ecosystem Simulator (LPJ-GUESS) dynamic vegetation model (Smith et al., 2014).


### CanESM5

The CanESM5 model (Swart et al., 2019) uses the CLASS-CTEM3.6 land surface model and the CanAM5 atmospheric model at T63 spectral resolution (approx. 2.8 degrees) and with 49 vertical levels. These are coupled every three hours to a customized version of the NEMO 3.4.1 model, with 45 vertical levels and a

nominal 1 degree horizontal resolution on the tripolar ORCA1 grid. CanESM5 uses the LIM2 sea ice model embedded within NEMO and discretized on the same grid, using a single thickness category each for ice and snow, and an elastic-viscous plastic rheology (Madec, 2016; Fichefet and Morales Maqueda, 1997). There is no penetration of insolation through snow, and there is no representation of lateral melt. Sea-ice dynamics and thermodynamics are computed within LIM2, while radiative fluxes are computed within CanAM5 and passed to

LIM2 via the coupler (Swart et al., 2019).

Radiation is calculated using the radiation scheme from CanAM5, as described in von Salzen (2013). Albedos for the bare ice and snow are computed for the 4 wavelength intervals used by the atmospheric model (von Salzen et al, 2013). The bare-ice and melt pond albedos are based on the parameterizations of Ebert and Curry (1993). The snow albedos are computed using the same parameterization used over land which accounts for

snow grain growth and the presence of black carbon deposited from the atmosphere.

### CESM2-CAM and CESM2-WACCM

The CESM2 models are built from the CESM2.1 configuration described in Danabasoglu et al. (2020). The CESM2-CAM configuration uses the CAM6 atmosphere model and the CLM5 land surface model at nominal 0.9

x 1.25 degree resolution, coupled to the POP 2.0.1 ocean and the Los Alamos sea ice model (CICE) at nominal 1 degree (gx1) resolution. The CESM2-WACCM configuration uses the WACCM6 high-top atmospheric component in place of CAM6.

The sea ice model component is based upon version 5.1.2 of the CICE sea ice model of Hunke et al. (2015). The CICE model uses a prognostic ice thickness distribution (ITD) with five thickness categories. The standard CICE

elastic–viscous–plastic (EVP) rheology is used for ice dynamics. The model uses mushy-layer thermodynamics and prognostic sea ice salinity (Turner et al., 2013) configured with 8 layers of ice and 3 of snow.

Radiation is calculated using the delta-Eddington scheme of Briegleb and Light (2007), with melt-ponds modelled on level, undeformed ice, as in Hunke et al. (2013).

### CSIRO_ARCCSS_ACCESS-CM2

The CSIRO_ARCCSS_ACCESS-CM2 uses atmosphere and sea mode components derived from the HadGEM3-GC3.1 models – including the MetUM atmosphere, using the Global Atmosphere 7.1 (GA7.1) configuration of Walters et al. (2017), and the Global Sea Ice 8.1 (GSI8.1) configuration of the CICE sea ice model described in Ridley et al. (2018). These are combined with version 2.5 of the CABLE land surface model and version 5.1 of

the MOM ocean model.

As for the HadGEM3-GC3.1 models, the sea ice model component is based upon version 5.1.2 of the CICE sea ice model (Hunke et al., 2015), using five thickness categories and the standard elastic–viscous–plastic (EVP) rheology. The model uses the Bitz and Lipscomb (1999) multi-layer vertical thermodynamics with 4 layers of ice and 1 of snow. Atmosphere-ice coupling is performed separately for each ice thickness category using the method of West et al., (2016).

Radiation is calculated using CICE's "default" CCSM3 scheme (see Hunke et al., 2015) which uses separate albedos for visible (< 700 nm) and near-infrared (> 700 nm) wavelengths for both bare ice and snow. The scheme has been ported into JULES where the surface exchanges are calculated. Prognostic melt ponds are included using the CICE topographic melt pond formulation of Flocco et al. (2012). Melt pond evolution is calculated in CICE with pond fraction and depth being passed through the coupler for use within the surface exchange scheme (albedo).

GFDL-CM4 and GFDL-ESM4

The GFDL-CM4 and GFDL-ESM4 models use the AM4.0 atmosphere/land model as documented in Zhao et al. (2018) with 100km horizontal resolution. CM4 has 33 vertical atmospheric levels whereas ESM4 has 49 vertical levels. These are coupled to the ocean and sea ice models MOM6 and SIS2, which are run under the OM4.0 configurations described in Adcroft et al. (2019). CM4 and ESM4 use nominal 0.25 and 0.5 degree horizontal ocean and sea ice resolutions, respectively. Both models have 75 vertical ocean layers. The CM4 model is documented in Held et al. (2019).

The SIS2.0 sea ice model has four sea ice layers and one snow layer and uses thermodynamics similar to Bitz and Lipscomb (1999), except that brine has the heat capacity of seawater rather than that of sea ice. The model uses the shortwave radiative transfer method of Briegleb and Light (2007), using the same code as the Community Sea Ice model CICE4.1 (Hunke & Lipscomb, 2010). The thermodynamic solver couples ice temperatures implicitly to the atmosphere via a surface skin temperature calculation (Winton, 2000). Ice salinities in each ice layer are prescribed as in CICE4.1. The model uses a constant coupling coefficient for basal ice-ocean heat flux of 240 $W/m^2/K$, and does not include a lateral melting scheme. SIS2.0 has a prognostic ice thickness distribution (ITD) with five ice thickness categories that are managed using the Lagrangian scheme of Bitz et al. (2001). Ice dynamics are computed using a C-grid horizontal stencil and an elastic-viscous-plastic rheology following Bouillon et al. (2009). Sea ice strength is calculated based on gridcell-averaged thickness and concentration following the formulation of Hibler (1979).

HadGEM3-GC31-LL and HadGEM3-GC31-MM

The HadGEM3-GC31 models use the Global Coupled configuration 3.1 (GC3.1) of the HadGEM3 model (Williams et al., 2017). This model comprises the MetUM atmosphere and JULES land models using the Global Atmosphere 7.1 (GA7.1) and Global Land 7.0 (GL7.0) configurations (Walters et al., 2017). These are coupled to the NEMO ocean and CICE sea ice models using the Global Ocean 6.0 (GO6.0, Storkey et al., 2018) and Global Sea Ice 8.1 (GSI8.1, Ridley et al., 2018).

The HadGEM3-GC31-MM model horizontal resolution is N216 for the atmosphere and land  and 0.25 degree (ORCA025) for the ocean and sea ice, and the HadGEM3-GC31-LL model horizontal resolution is N96 for the

atmosphere and land and 1 degree (ORCA1) for the ocean and sea ice. Both HadGEM3-GC31 models use 85 vertical levels in the atmosphere and the ocean components use 75 vertical levels.

The sea ice model component is based upon CICE version 5.1.2 (Hunke et al., 2015), with a prognostic ice thickness distribution (ITD) with five thickness categories. The standard CICE elastic–viscous–plastic (EVP) rheology is used for ice dynamics (Hunke et al., 2015). The model uses the Bitz and Lipscomb (1999) multi-layer

vertical thermodynamics with 4 layers of ice and 1 of snow.

Atmosphere-ice coupling is performed separately for each ice thickness category using the method of West et al., (2016). Radiation is calculated using CICE's "default" CCSM3 scheme (Hunke et al., 2015). Prognostic melt ponds are included using the CICE topographic melt pond formulation of Flocco et al. (2010).

There are several minor differences between the LL and MM resolution versions of HadGEM3-GC3.1, which are

displayed in Table 2 of Roberts et al. (2019).

IPSL-CM6A-LR

IPSL-CM6A-LR is the 6th version of the IPSL-CM model (Boucher et al., 2020). IPSL-CM6A-LR couples the atmospheric model LMDZ6 (Hourdin et al., 2019) to NEMO3.6 (Madec et al., 2017), including LIM3.6 as a sea

ice component (Vancoppenolle et al., 2009, Rousset et al., 2015).

Horizontal resolution is 144 x 142 grid points for the atmosphere (157 km), and nominal 1 degree (eORCA1) for both the ocean and sea ice. Sea ice model resolution in the Arctic is about 50 km. The atmospheric model has 79 vertical layers, with a model top at ~80 km, whereas the ocean has 75 vertical levels.

The sea ice component is based upon version 3.6 of  the Louvain-la-Neuve Ice Model (LIM3.6). LIM3.6 is a

multi-category halo-thermodynamic dynamic sea ice model embedded the NEMO environment (including the horizontal grid). It is based on the Arctic Ice Dynamics Joint EXperiment (AIDJEX) framework (Coon et al., 1974), combining the ice thickness distribution framework, the conservation of horizontal momentum, an elastic-viscous plastic rheology, and energy-conserving halo-thermodynamics, combining the Bitz and Lipscomb (1999) multi-layer thermodynamics (2/1 vertical levels in ice/snow) and a semi-empirical

parameterization of brine drainage (Vancoppenolle et al., 2009). The surface albedo is empirically specified as a function of the ice surface temperature, ice thickness, snow depth, and cloudiness. The radiation absorbed by the ice follows Beer's law and does not infiltrate into the snow. Melt ponds are crudely accounted for, by reducing the albedo by 0.06 when ice is melting from the surface. LIM3 does not account for the lateral melting.

MRI-ESM2

The MRI-ESM2 model (Yukimoto et al., 2019) uses version 3.5 of the MRI-AGCM atmosphere/land model at TL159 (approximately 120 km) horizontal resolution with 80 vertical levels. The ocean and sea ice are modelled using version 4.4 of the Meteorological Research Institute Community Ocean Model (MRI.COM; Tsujino et al., 2017) using a resolution of 1 degree zonally and 0.3–0.5 degree meridionally and a tri-polar grid north of 64ºN

with a 30–50 km resolution in the Arctic Ocean.

The sea ice model component uses a prognostic ITD with 5 thickness categories and uses the elastic–viscous–plastic rheology of Hunke and Dukowicz (1997). Sea ice thermodynamics are performed following Mellor and Kantha (1989) using 1 layer of ice and a zero-layer snow scheme. Radiation is performed using the 'default' CICE CCSM3 described in Hunke et al. (2015). Lateral melting is not modelled in MRI-COMv4 and so lateral

melting is considered part of basal melting. In addition, ice creation from the supercooled water within a grid is all counted as basal growth, whereas frazil ice growth is not defined.

NorESM2-LM and NorESM2-MM

NorESM2 is built on the structure and many of the components in CESM2.1 as described in Danabasoglu et al.
(2020), with modifications as detailed by Seland et al (2020). NorESM2-LM has an atmospheric resolution of 2 degrees, and NorESM2-MM 1 degree.

The atmospheric model in NorESM2 (NorCAM6) is based on CAM6, but with modified aerosol-radiation-cloud interaction (Kirkevåg et al. 2013; 2018) as well as changes in energy, momentum and flux. The land model CLM5 is close to the version in CESM2.1, and the river transport model Mosart is identical to the version found in
CESM2.1. The ocean model BLOM and ocean-biogeochemistry model, HAMOCC, are upgraded versions of the components found in NorESM1 (Bentsen et al, Iversen et al, 2013). The ocean model and ocean– sea ice configuration are described in detail in Bentsen et al (in prep).

The sea ice model component is based upon version 5.1.2 of the CICE sea ice model of Hunke et al. (2015). The NorESM2-specific changes includes effects of wind drift of snow into ocean following Lecomte et al (2013) as
described in Bentsen et al. (in prep), and a zenith angle-based time average of albedo.

The CICE model uses a prognostic ice thickness distribution (ITD) with five thickness categories. The standard CICE elastic–viscous–plastic (EVP) rheology is used for ice dynamics (Hunke et al., 2015). The model uses mushy-layer thermodynamics and prognostic sea ice salinity (Turner et al., 2013) configured with 8 layers of ice and 3 of snow.
Radiation is calculated using the delta-Eddington scheme of Briegleb and Light (2007). Melt ponds are modelled on level, undeformed ice, as in Hunke et al. (2013).

UKESM1-0-LL

The UKESM1-0-LL model is the first version of the United Kingdom Earth System Model. UKESM1-0-LL uses
the coupled climate model HadGEM3-GC31-LL as its physical core, with the following components interactively coupled: terrestrial carbon and nitrogen cycles, including dynamic vegetation and representation of agricultural land use change; ocean biogeochemistry (BGC) with prognostic diatom and non-diatom concentrations and a unified troposphere-stratosphere chemistry model, tightly coupled to a multi-species modal aerosol scheme.

The model resolution – both vertical and horizontal – and details of the sea ice and ocean components of
UKESM1-0-LL are otherwise identical to that of HadGEM3-GC31-LL described above.

The UKESM1-0-LL model is described further in Sellar et al. (2019).

**Author Contributions**

AK and EB defined the scope of the study. AK performed the model comparisons. AK wrote the paper, in collaboration with EB. LP provided one of the figures. DB, MB, DD, DF FM, LP, DS and NS provided text and comments. AK, DB, JB-D, MB, SD, DD, S O'F, LP, J M-R, DS, NS, TT, HT, MV and KW provided processed data for the study.

**Acknowledgements**

We thank Chris Derksen, Till Wagner and an anonymous referee for editing and reviewing our paper, and for their very useful suggestions which have helped to improve the manuscript and figures.

Ann Keen and Ed Blockley were supported by the Joint UK BEIS/Defra Met Office Hadley Centre Climate Programme (GA01101).

Ann Keen and Ed Blockley were supported by the European Union's Horizon 2020 Research & Innovation programme through grant agreement no 727862 APPLICATE.

Leandro Ponsoni was funded by the APPLICATE project until September 2019 and is now funded by the Fonds de la Recherche Scientifique (FNRS).

David Docquier was funded by the EU Horizon 2020 PRIMAVERA project, grant agreement no. 641727 (until September 2019), and is currently funded by the EU Horizon 2020 OSeaIce project, under the Marie Sklodowska-Curie grant agreement no. 834493.

David Schroeder was funded under the ACSIS and UKESM program (UK Natural Environment Research Council)

Jens Boldingh Debernard was supported by the Research Council of Norway through the projects INES (270061) and KeyClim (295046).

David Bailey is supported by the National Center for Atmospheric Research (NCAR), which is a major facility sponsored by the NSF under Cooperative Agreement No. 1852977.

The ACCESS-CM2 CMIP6 submission was jointly funded through CSIRO and the Earth Systems and Climate Change Hub of the Australian Government's National Environmental Science Program, with support from the Australian Research Council Centre of Excellence for Climate System Science. The ACCESS model simulations and data publication were supported by the National Computational Infrastructure (NCI).

We thank Dorotea Iovino for providing useful comments on the manuscript, and we thank all those who have worked to develop the CMIP6 models and run the model integrations.

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

**Tables**

| Model Name | Modelling Centre | No of integrations | | Notes |
|---|---|---|---|---|
| | | Hist | SSP58.5 | |
| HadGEM3-GC31-LL | Met Office | 4 | 4 | |
| HadGEM3-GC31-MM | Met Office | 4 | 4 | |
| UKESM1-0-LL | Met Office | 12 | 5 | |
| EC-Earth3 | UCLouvain/AEMET | 1 | 1 | No explicit lateral melt |
| EC-Earth3-Veg | UCLouvain/SMHI | 1 | 1 | No explicit lateral melt, dynamics term missing. |
| MRI-ESM2 | MRI | 5 | 1 | No explicit lateral melt or frazil ice formation |
| CESM2-CAM | NCAR | 11 | 2 | |
| CESM2-WACCM | NCAR | 3 | 2 | |
| GFDL-CM4 | GFDL | 1 | 1 | No explicit lateral melt |
| GFDL-ESM4 | GFDL | 1 | 1 | No explicit lateral melt |
| CSIRO_ARCCSS_ACCESS-CM2 | CSIRO | 1 | 1 | |
| NorESM2-LM | Met Norway | 3 | 1 | |
| NorESM2-MM | Met Norway | 1 | 1 | |
| CanESM5 | ECCC | 3 | 3 | Missing terms: frazil, lateral melt, evapsubl, dynamics. |
| IPSL_ | IPSL | 32 | 6 | No explicit lateral melt |
| NEMOCICE_CORE_default | CPOM | 1 | | Forced ocean-ice model (so no scenario data) |
| NEMOCICE_CORE_CPOM-CICE | CPOM | 1 | | |
| NEMOCICE_DFS5.2_CPOM-CICE | CPOM | 1 | | |

***Table 1****: List of models and modelling centres participating in this study. Where two modelling centres are shown, the 1st analysed the model outputs, and the 2nd performed the model integrations.*




| Components, configurations, resolution: | UKESM1-0-LL | HadGEM3-GC31-LL | HadGEM3-GC31-MM | AEMET_EC-Earth3 | SMHI_EC-Earth3-Veg | ACCESS-CM2 | MRI-ESM2 |
|---|---|---|---|---|---|---|---|
| Sea ice model component (configuration) | CICE 5.1.2 (GSI8.1) | CICE 5.1.2 (GSI8.1) | CICE 5.1.2 (GSI8.1) | LIM3 | LIM3 | CICE 5.1.2 (GSI8.1) | MRI. COM4.4 |
| Sea ice model resolution | 1° (ORCA1) | 1° (ORCA1) | 0.25° (ORCA025) | 1° (ORCA1) | 1° (ORCA1) | 1° | 1° (lon) x 0.3-0.5° (lat) |
| Ocean model component (configuration) | NEMO3.6 (GO6) | NEMO3.6 (GO6) | NEMO3.6 (GO6) | NEMO3.6 | NEMO3.6 | MOM5.1 (ACCESS-OM) | MRI. COM4.4 |
| Ocean model resolution | 1° (ORCA1) | 1° (ORCA1) | 0.25° (ORCA025) | 1° (ORCA1) | 1° (ORCA1) | 1° | 1° (lon) x 0.3-0.5° (lat) |
| Atmosphere model component (configuration) | MetUM (GA7.1) | MetUM (GA7.1) | MetUM (GA7.1) | IFS (cycle36r4) | IFS (cycle36r4) | MetUM (GA7.1) | MRI-AGCM 3.5 |
| Atmosphere model resolution | 135 km (N96) | 135 km (N96) | 60 km (N216) | 80 km (T255L91) | 80 km (T255L91) | 135 km (N96) | 120 km (TL159) |
| **Sea Ice model specifics** | | | | | | | |
| Rheology | EVP | EVP | EVP | EVP | EVP | EVP | EVP |
| Ice Thickness Distribution (ITD) | Prognostic | Prognostic | Prognostic | Prognostic | Prognostic | Prognostic | Prognostic |
| No of thickness categories | 5 | 5 | 5 | 5 | 5 | 5 | 5 |
| Radiation scheme | Dual band (CCSM3) | Dual band (CCSM3) | Dual band (CCSM3) | Broadband | Broadband | Dual band (CCSM3) | Dual band (CCSM3) |
| Penetration of SW into ocean | No | No | No | Yes | Yes | No | Yes |
| Melt ponds | Prognostic (F12) | Prognostic (F12) | Prognostic (F12) | Prescribed albedo reduction | Prescribed albedo reduction | Prognostic (F12) | Parameterised |
| Thermodynamics | BL99 | BL99 | BL99 | BL99 + prognostic salinity profile | BL99 + prognostic salinity profile | BL99 | MK89 |
| No of ice (snow) layers | 4(1) | 4(1) | 4(1) | 2(1) | 2(1) | 4(1) | 1(0) |
| Minimum lead fraction | None | None | None | 0.003 | 0.003 | None | None |
| Minimum frazil thickness | 5 cm | 5 cm | 5 cm | 10 cm | 10 cm | 5 cm | - |
| **Coupling, time-stepping** | | | | | | | |
| Sea ice model timestep | 30 mins | 30 mins | 20 mins | 45 mins | 45 mins | 30 mins | 30 mins |
| Ice-ocean coupling frequency | 30 mins | 30 mins | 20 mins | 45 mins | 45 mins | 30 mins | 30 mins |
| Ice-atmosphere coupling frequency | 20 mins (180 mins) | 20 mins (180 mins) | 20 mins (60 mins) | 45 mins | 45 mins | 20 mins (180 mins) | 60 mins |


*Table 2*: Relevant information for all the CMIP6 models used in this study including a summary of the model subcomponents and resolution, along with details of several key physical aspects of the sea ice subcomponents. For thermodynamics: T13 denotes the mushy-layer scheme of Turner et al. (2013), whilst BL99 and MK89 denote fixed salinity-profile schemes of Bitz and Lipscomb (1999) and Mellor and Kantha (1989), respectively. For prognostic melt-ponds: F12 denotes the 'topographic' scheme of Flocco et al. (2012), whilst H13 denotes the 'level ice' scheme of Hunke et al. (2013). For rheology: EVP is the Elastic-Viscous-Plastic scheme of Hunke and Dukowicz (1997).


| Components, configurations, resolution: | GFDL-CM4 | GFDL-ESM4 | CanESM5 | CESM2-CAM | CESM2-WACCAM | NorESM-LL | NorESM-MM | IPSL-CM6A-LR |
|---|---|---|---|---|---|---|---|---|
| Sea ice model component (configuration) | SIS2 | SIS2 | LIM2 | CICE 5.2.1 | CICE 5.2.1 | CICE 5.2.1 | CICE 5.2.1 | LIM3 |
| Sea ice model resolution | 0.25° | 0.5° | 1° (ORCA1) | 1° (gx1) | 1° (gx1) | 1° (tn1v4) | 1° (tn1v4) | 1° |
| Ocean model component (configuration) | MOM 6 (OM4.0) | MOM 6 (OM4.0) | NEMO 3.4.1 | POP 2.0.1 | POP 2.0.1 | BLOM | BLOM | NEMO 3.6 |
| Ocean model resolution | 0.25° | 0.5° | 1° (ORCA1) | 1° (gx1) | 1° (gx1) | 1° (tn1v4) | 1° (tn1v4) | 1° |
| Atmosphere model component (configuration) | GFDL-AM (AM4.0) | GFDL-AM (AM4.0) | CanAM | CAM 6 | CAM 6 | NorCAM6 | NorCAM6 | LMDZ6 |
| Atmosphere model resolution | 100 km | 100 km | 2.8° (T63) | 0.9° (lon) x 1.25° (lat) | 0.9° (lon) x 1.25° (lat) | 1.9° (lon) x 2.5° (lat) | 0.9° (lon) x 1.25° (lat) | 2.5° x 1.3° (144x142) |
| **Sea Ice model specifics** | | | | | | | | |
| Rheology | EVP | EVP | EVP | EVP | EVP | EVP | EVP | EVP |
| Ice Thickness Distribution (ITD) | Prognostic | Prognostic | Diagnostic | Prognostic | Prognostic | Prognostic | Prognostic | Prognostic |
| No of thickness categories | 5 | 5 | 1 | 5 | 5 | 5 | 5 | 5 |
| Radiation scheme | delta-Eddington | delta-Eddington | Multi-band | delta-Eddington | delta-Eddington | delta-Eddington | delta-Eddington | broadband |
| Penetration of SW into ocean | Yes | Yes | Yes | Yes | Yes | Yes | Yes | Yes |
| Melt ponds | Parameterised | Parameterised | Parameterised | Prognostic (H13) | Prognostic (H13) | Prognostic (H13) | Prognostic (H13) | Prescribed albedo reduction |
| Thermodynamics | BL99 | BL99 | Semtner 0-layer | T13 | T13 | T13 | T13 | BL99 + prognostic salinity profile |
| No of ice (snow) layers | 4(1) | 4(1) | N/A | 8(3) | 8(3) | 8(3) | 8(3) | 2(1) |
| Minimum lead fraction | None | None | 1.0e-6 | None | None | None | None | 0.003 |
| Minimum frazil thickness | None | None | 5 cm | 5 cm | 5 cm | 5 cm | 5cm | 10 cm |
| **Coupling, time-stepping** | | | | | | | | |
| Sea ice model timestep | 20 mins | 20 mins | 60mins | 30 mins | 30 mins | 30 mins | 30 mins | 90 mins |
| Ice-ocean coupling frequency | 60 mins | 60 mins | 60 mins | 30 mins | 30 mins | 30 mins | 30 mins | 90 mins |
| Ice-atmosphere coupling frequency | 20 mins | 20 mins | 180 mins | 60 mins | 60 mins | 30 mins | 30 mins | 90 mins |




| | CPOM_NEMOCICE_ CORE_default | CPOM_NEMOCICE_ CORE_CPOM-CICE | CPOM_NEMOCICE_ DFS5.2_CPOM-CICE | HadGEM3-GC3.1-LL |
|---|---|---|---|---|
| **Atmospheric forcing:** | | | | |
| Dataset | CORE II | CORE II | DFS5.2 | MetUM coupled |
| Resolution | ~200km | ~200km | ~80km | ~135km (N96) |
| Frequency | 6 hourly | 6 hourly | 6 hourly | 3 hourly |
| **Sea ice model physics:** | | | | |
| Rheology | EVP | EAP | EAP | EVP |
| Sea ice conductivity | "default" (Maykut and Untersteiner, 1971) | "bubbly brine" (Pringle et al., 2007) | "bubbly brine" (Pringle et al., 2007) | "default" (Maykut and Untersteiner, 1971) |
| Sea ice emissivity | 0.95 | 0.976 | 0.976 | 0.95 |
| Melt pond max fraction (rfracmax) | 85% | 50% | 50% | 85% |
| Blown snow scheme | None | Schroder et al. (2019) | Schroder et al. (2019) | None |

*Table 3*: Relevant information describing the CPOM NEMO-CICE forced models used in this study. Including details of the atmospheric forcing datasets used and the changes in the CICE sea ice physics used in the different simulations. Also included is the HadGEM3-GC3.1-LL model for reference. All physics options not included in this table – including those shown in Table 2 above – are identical to the HadGEM3-GC3.1-LL model. CORE II is the CORE2 surface forcing data set of Large and Yeager (2009), whilst DFS5.2 is the Drakkar forcing set of Dussin et al. (2016). For rheology: EVP is the Elastic-Viscous-Plastic scheme of Hunke and Dukowicz (1997); EAP is the Elastic-Anisotropic-Plastic scheme of Tsamados et al (2013). The melt-pond max fraction term (rfracmax) determines the maximum fraction of melt-water added to the ponds within the Flocco et al. (2012) 'topographic' prognostic melt-pond scheme.



| Sea ice configuration | Thermodynamics | Melt ponds | Minimum frazil thickness |
|---|---|---|---|
| **CICE 5.1.2** *(CICEa)* | T13 | Prognostic H13 | 5cm |
| **CICE5.1.2 (GSI8.1)** *(CICEb)* | BL99 | Prognostic F12 | 5cm |
| **LIM3** | BL99 | Prescribed albedo reduction | 10cm |
| **LIM2** | 0-Layer | Parameterised | 5cm |
| **SIS2** | BL99 | Parameterised | None |
| **MRI.COM 4.4** *(COM4)* | MK89 | Parameterised | - |

*Table 4: Selected features of the sea ice components used by the CMIP6 models. References for each scheme are provided in Table 2.*










**Figures**

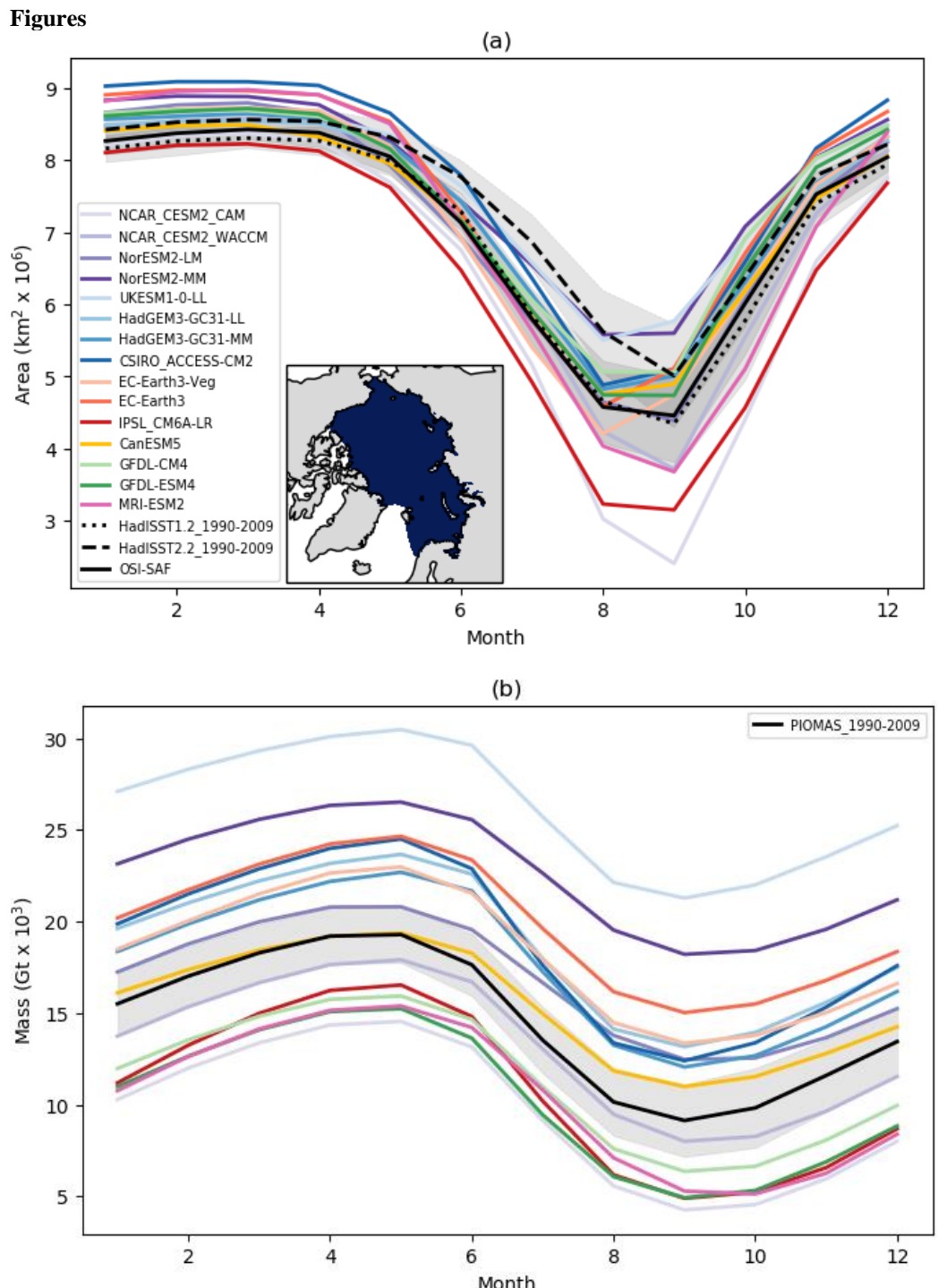

*Figure 1: Seasonal cycles of (a) ice area and (b) ice mass for the reference period 1990-2009, for the CMIP6 models. Where more than one model integration is available, the values are ensemble means. Also shown is data from the  (a) HadISST1.2 (Rayner et al., 2003), HadISST.2.2 (Titchner and Rayner, 2014) and OSI-SAF (OSI-SAF, 2017) observational datasets, and (b) PIOMAS reanalysis (Zhang and Rothrock, 2003) for the same period. The shaded regions show +/- 1 standard deviation in the monthly values. Data is summed over the analysis domain shown in (a), which is defined using the NSIDC Arctic regional masks (https://nsidc.org/data/polar-stereo/tools_masks.html#region_masks), where we include the following regions: Central Arctic Ocean, Beaufort Sea, Chukchi Sea,  East Siberian Sea, Laptev Sea, Kara Sea and Barents Sea.*

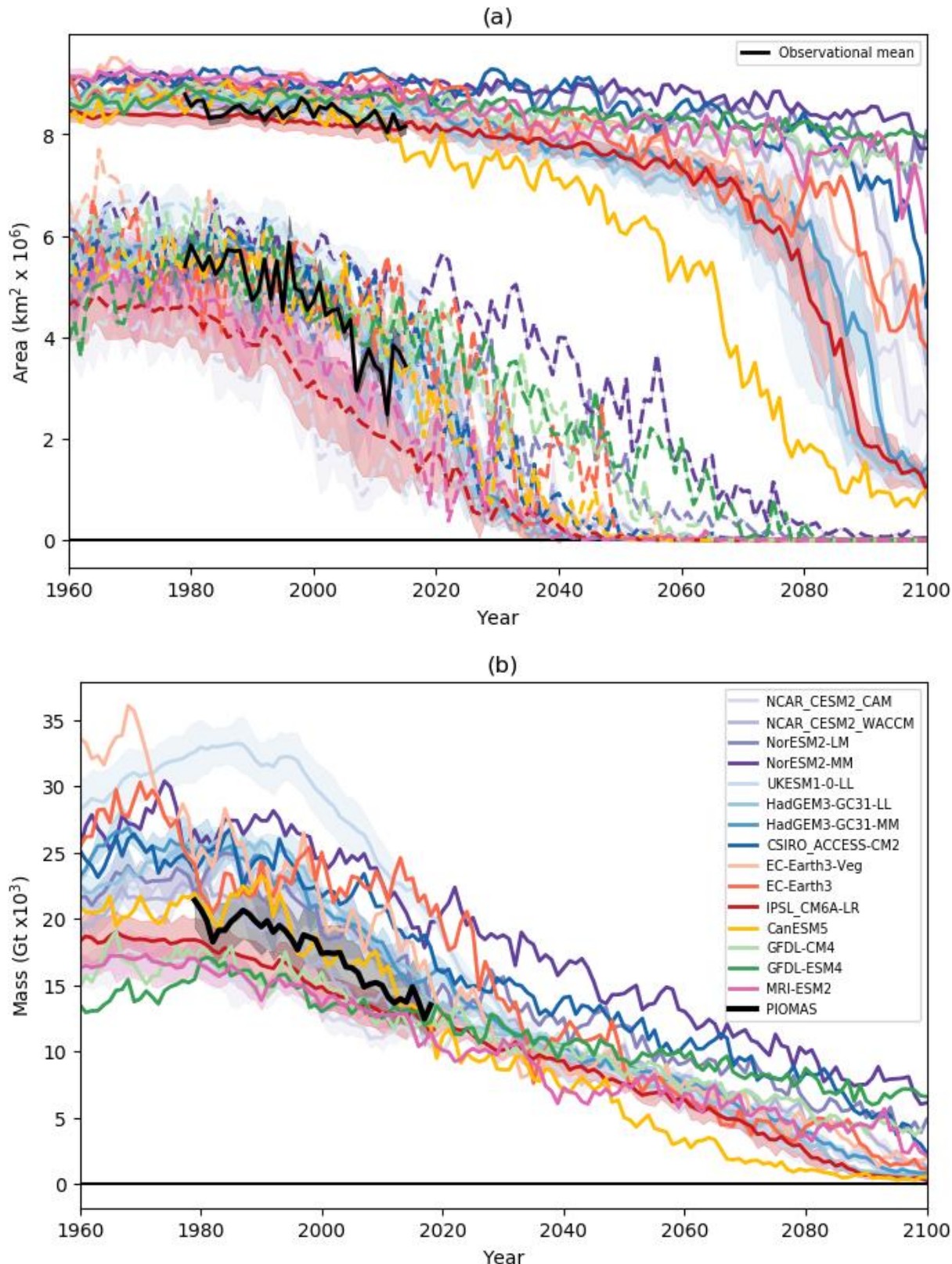


*Figure 2: Evolution of (a) March (solid lines) and September (dash lines) ice area and (b) March ice mass for the CMIP6 models. Also shown is (a) the mean and range for the for the HadISST1.2 (Rayner et al., 2003), HadISST.2.2 (Titchner and Rayner, 2014) and OSI-SAF (OSI-SAF, 2017) observational datasets, and (b) the PIOMAS reanalysis (Zhang and Rothrock, 2003) with uncertainty estimate*

*(Schweiger et al, 2011).*

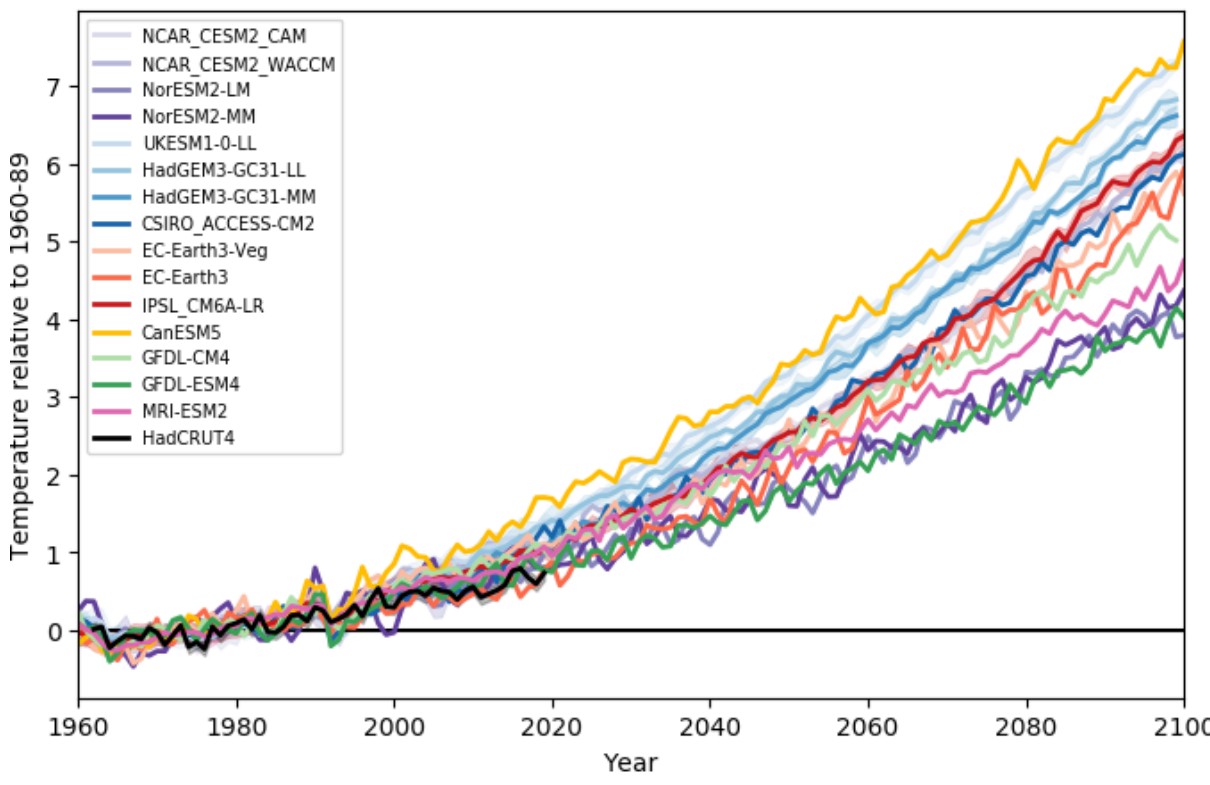


*Figure 3: Evolution of annual-mean global-mean near-surface temperature for the CMIP6 models and the HadCRUT4 observational dataset (Morice et al, 2012). Values are relative to the 1960-89 mean in each case.*


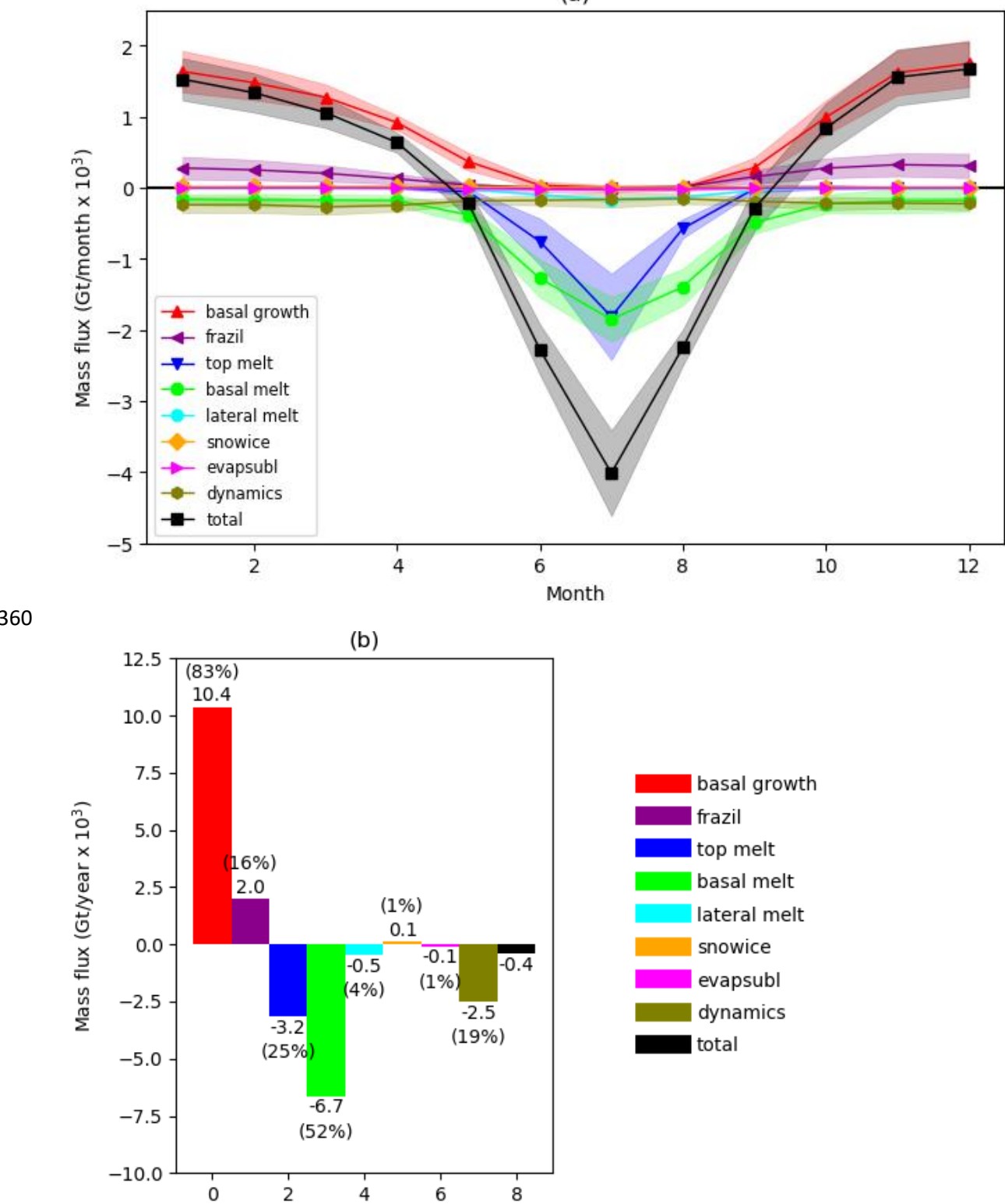


*Figure 4: Components of the Arctic sea ice mass budget for the multi-model mean. Values are summed over the region shown in Fig. 1, for the period 1960-89. For each budget component, values*

*are calculated by averaging the ensemble mean for each model with a non-zero value of the component. (a) Seasonal cycle of monthly mean values, (b) annual mean values. The shaded regions show +/- 1 standard deviation in the modelled values. Percentages are relative to the total annual mass of ice growth or loss.*


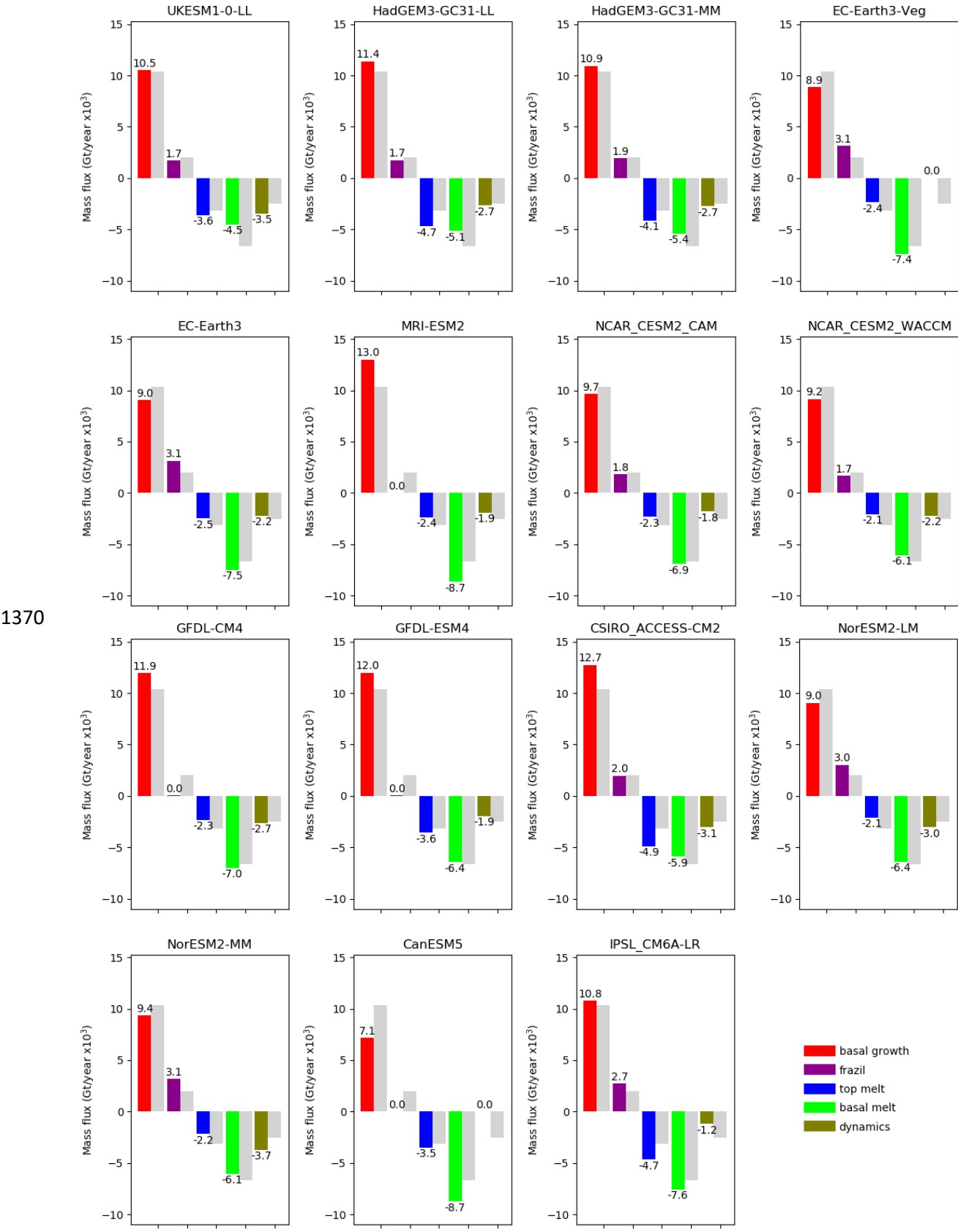

*Figure 5: Main components of the annual mean Arctic sea ice mass budget for each model. Values are summed over the region shown in Fig. 1, for the period 1960-89. The grey bars show the multi-model mean for each term for the same period.*

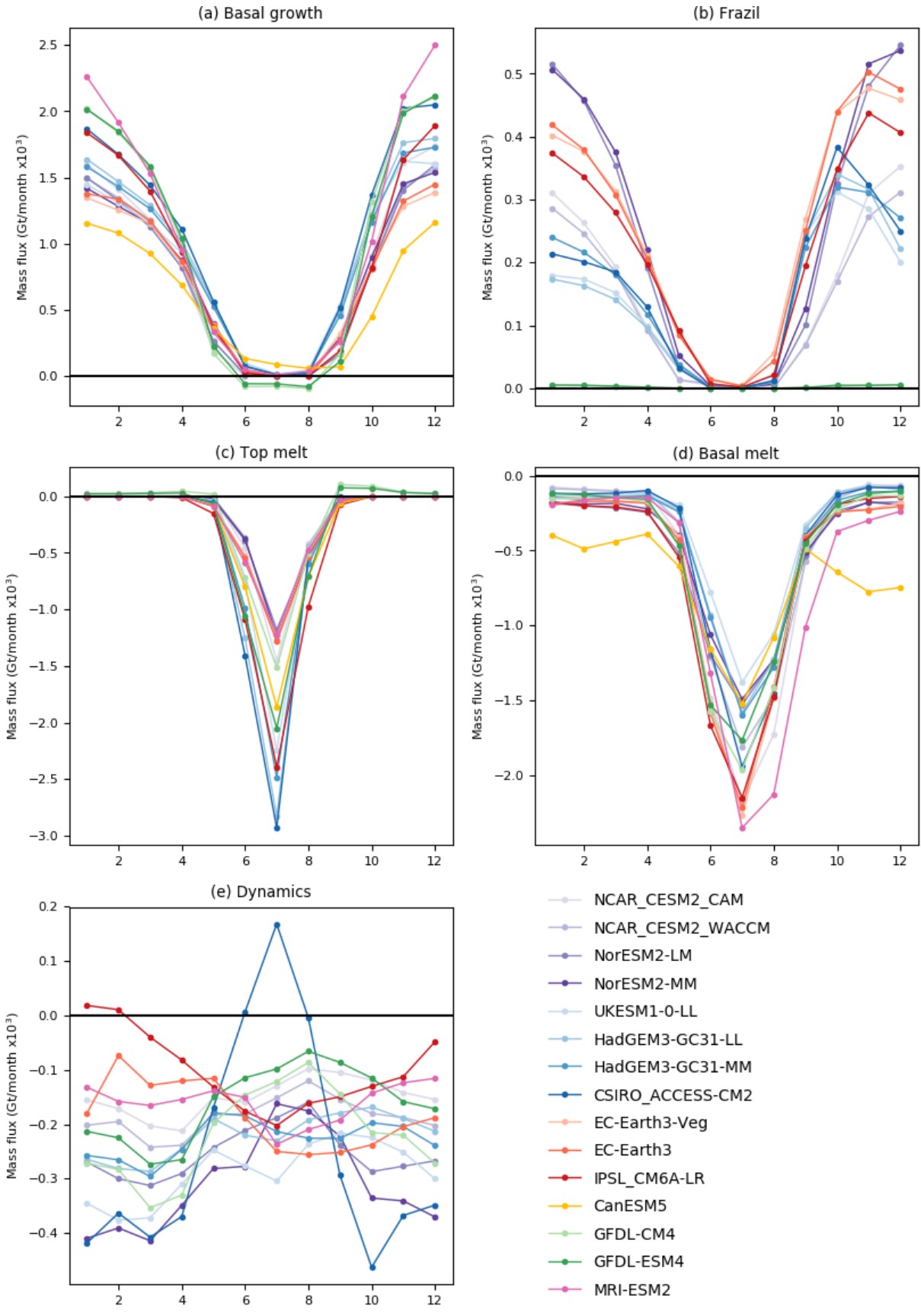


*Figure 6: Seasonal cycles of the main components of the Arctic sea ice mass budget. Values are summed over the region shown in Fig. 1, for the period 1960-89.*

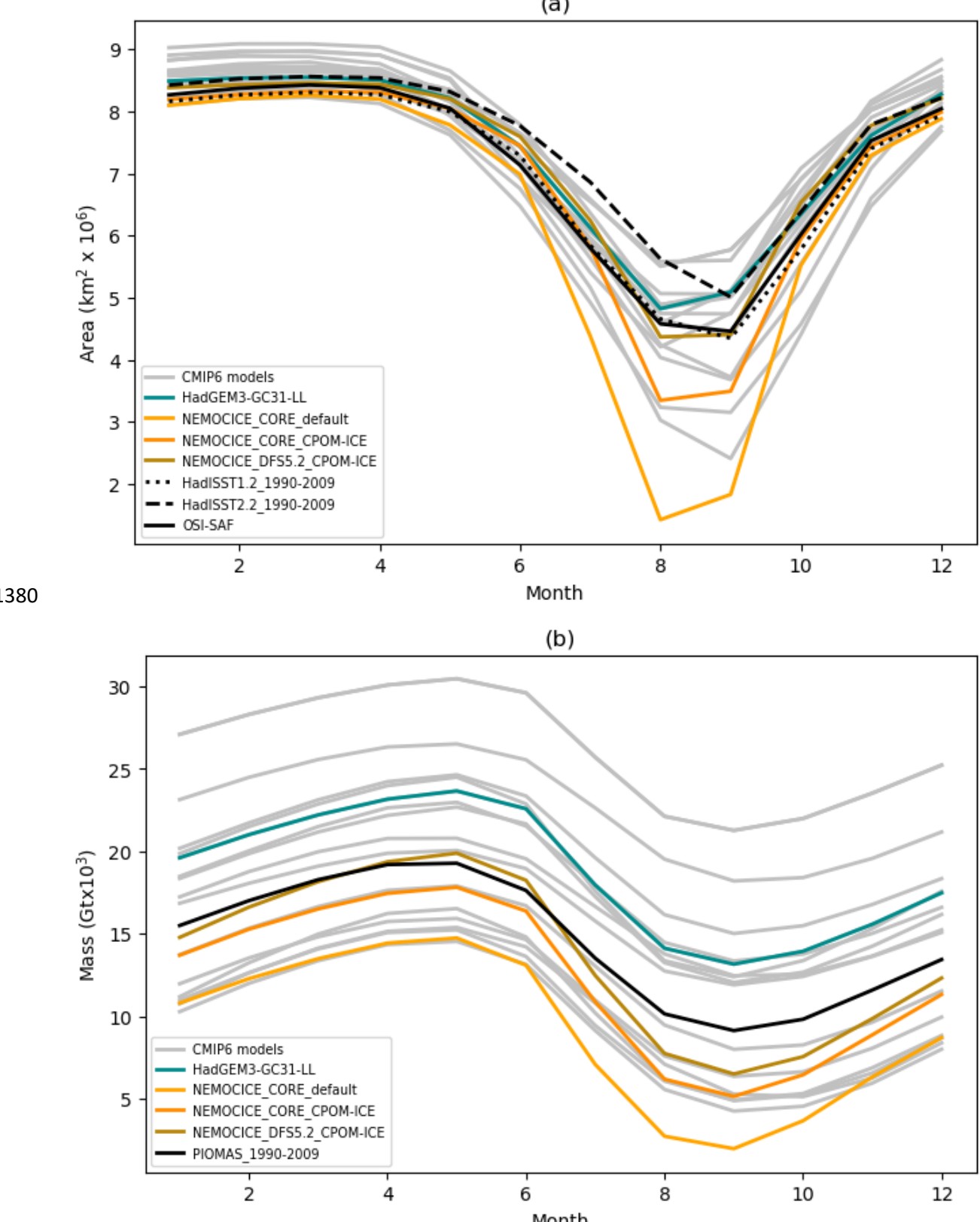

*Figure 7: Seasonal cycles of (a) ice area and (b) ice volume for the reference period 1990-2009 budget, for the forced ocean-ice runs (Table 3), plus HadGEM3-GC31-LL and the other CMIP6 models. Where more than one model integration is available, the values are ensemble means. Also shown is data for the same periods for (a) the HadISST1.2 (Rayner et al., 2003), HadISST.2.2 (Titchner and Rayner, 2014), and OSI-SAF (OSI-SAF, 2017) observational datasets, and (b) PIOMAS reanalysis (Zhang and Rothrock, 2003) for the same period.*

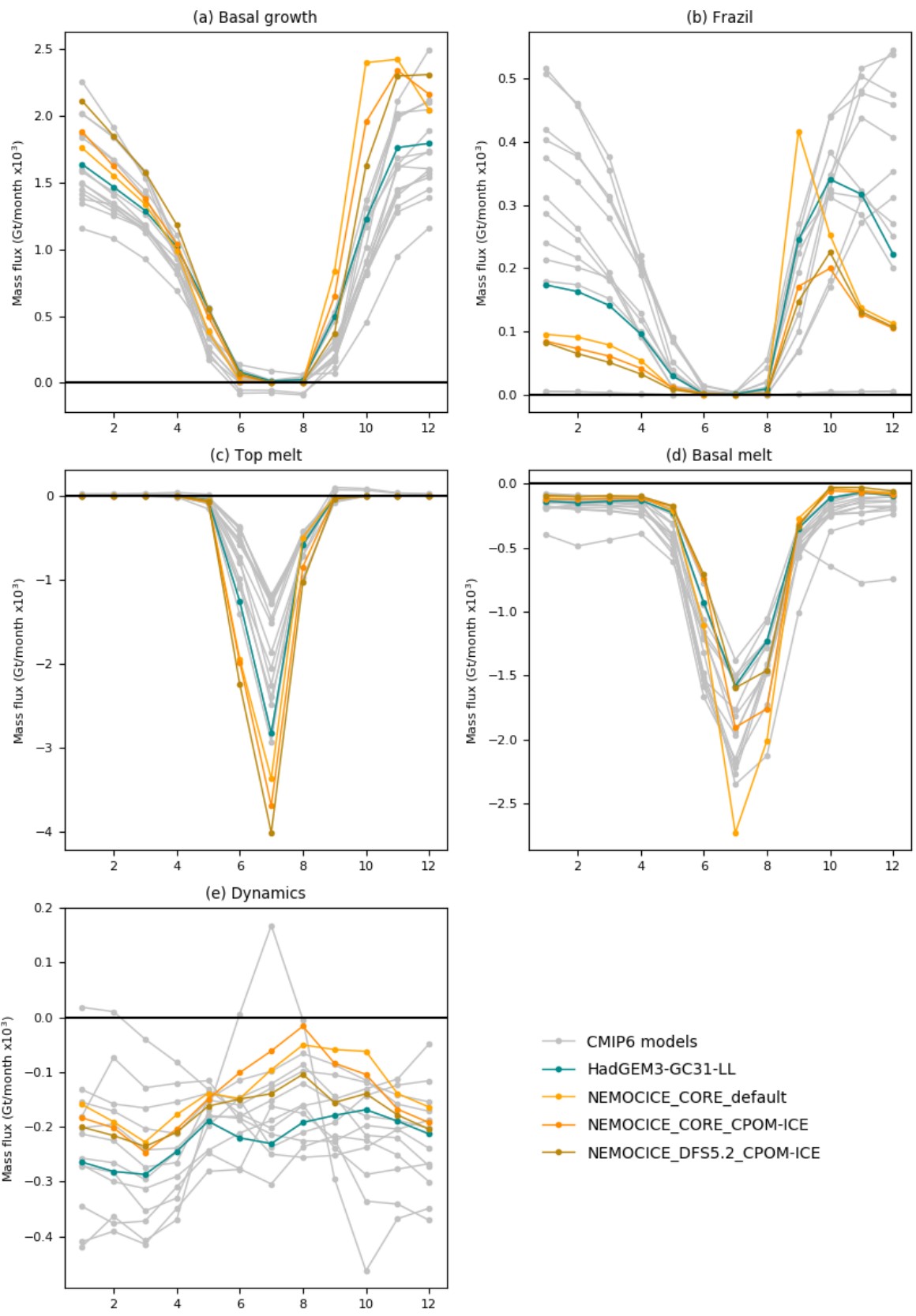

*Figure 8: Seasonal cycles of the main components of the Arctic sea ice mass budget. Values are summed over the region shown in Fig. 1, for the period 1960-89, for the forced ocean-ice runs (Table 3), plus HadGEM3-GC31-LL and the other CMIP6 models.*

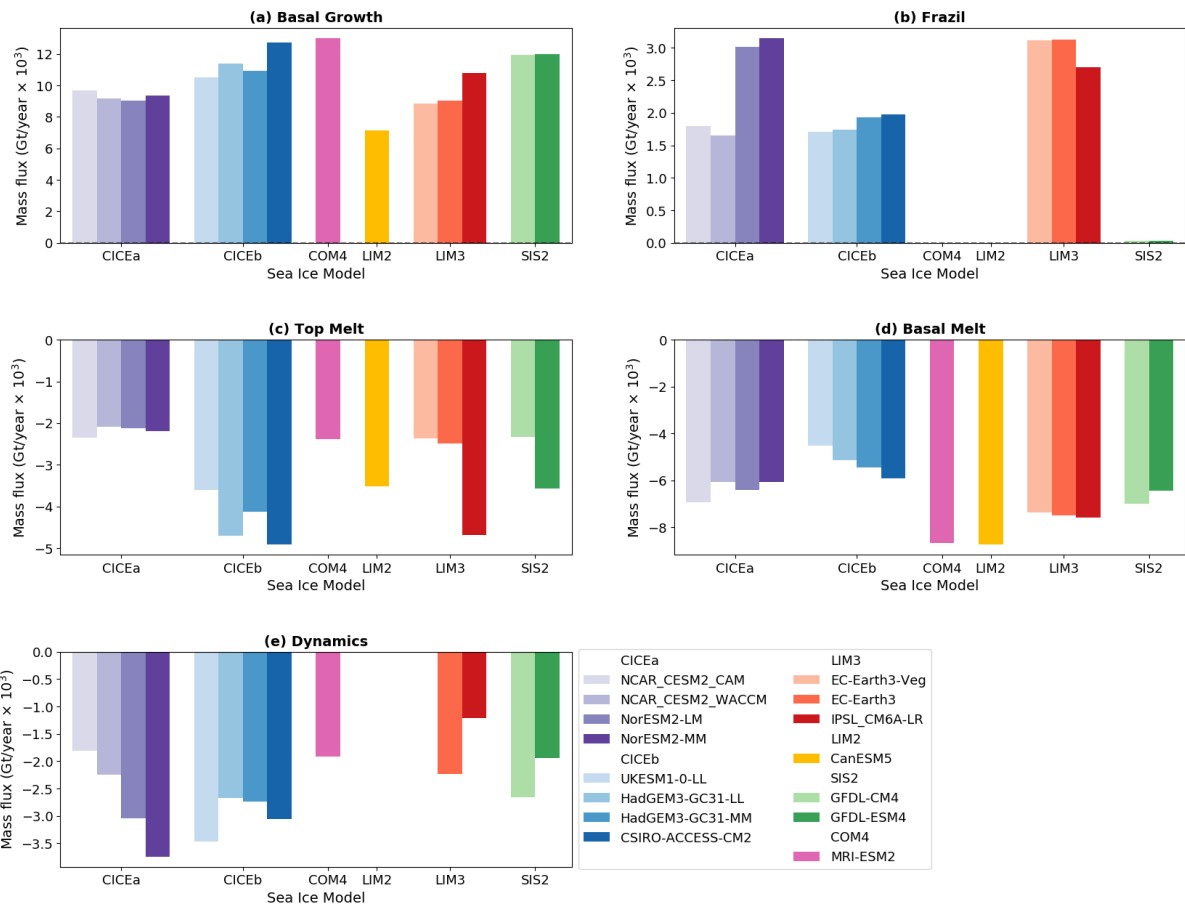


*Figure 9: Main components of the annual mean Arctic sea ice mass budget for each model, for the reference period 1960-89. Values are grouped by key features of the sea ice model as summarised in Table 4, and where  CICEa is CICE 5.1.2, CICEb is CICE 5.1.2 (GSI8.1), and COM4 is MRI_COM4.4.*


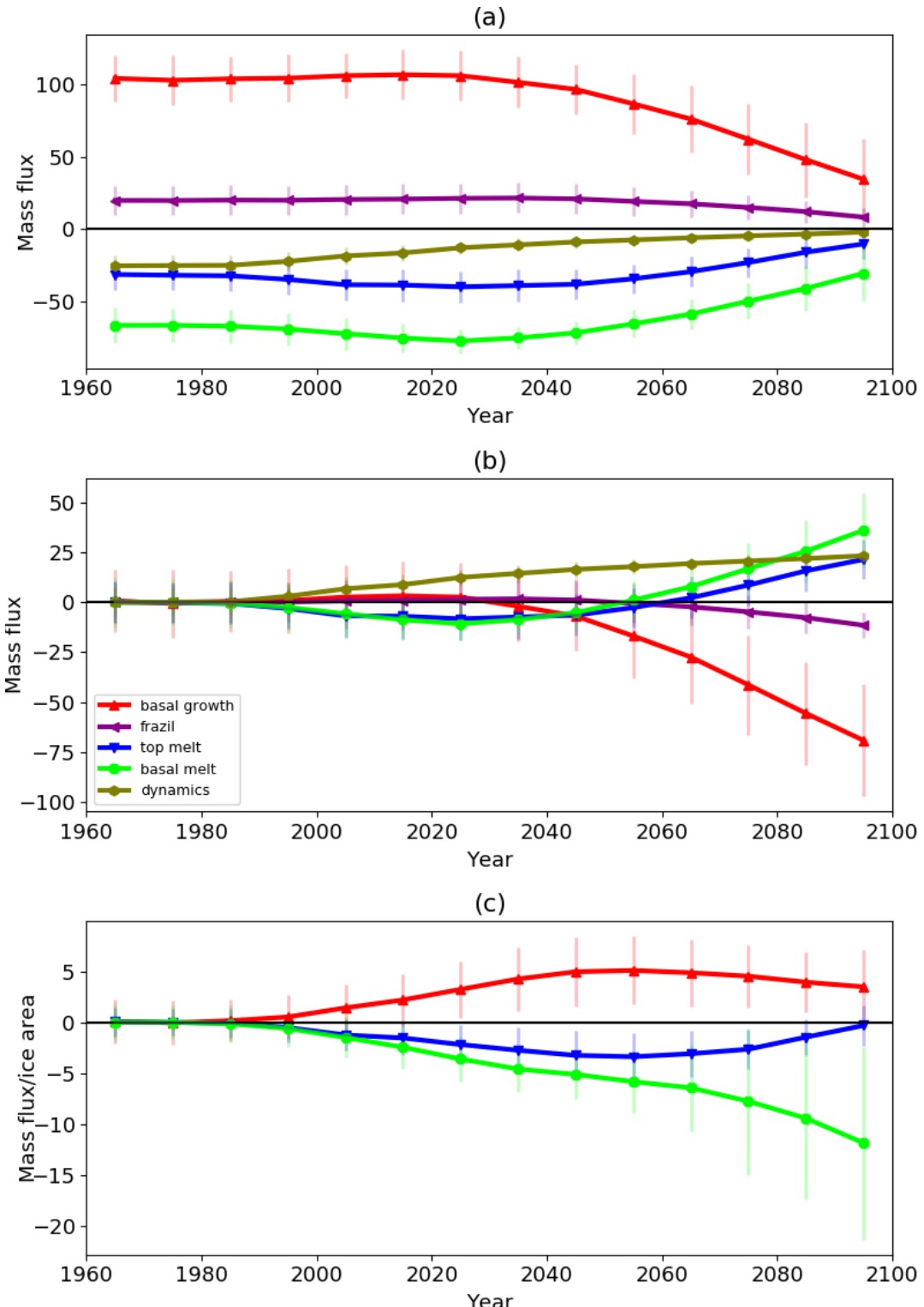

*Figure 10: (a) Evolution of decadal mean components of the main terms in the Arctic sea ice mass budget for the multi-model mean (Gt x10³ per decade) Values are summed over the region shown in Fig. 1. (b) Anomalies relative to 1960-89 for multi-model mean budget terms (Gt x10³ per decade) (c) Anomalies relative to 1960-80 per unit ice area for budget terms representing processes acting at the ice surface (Gt/decade per kmx10²). Error bars are +/- 1 standard deviation in the modelled values.*


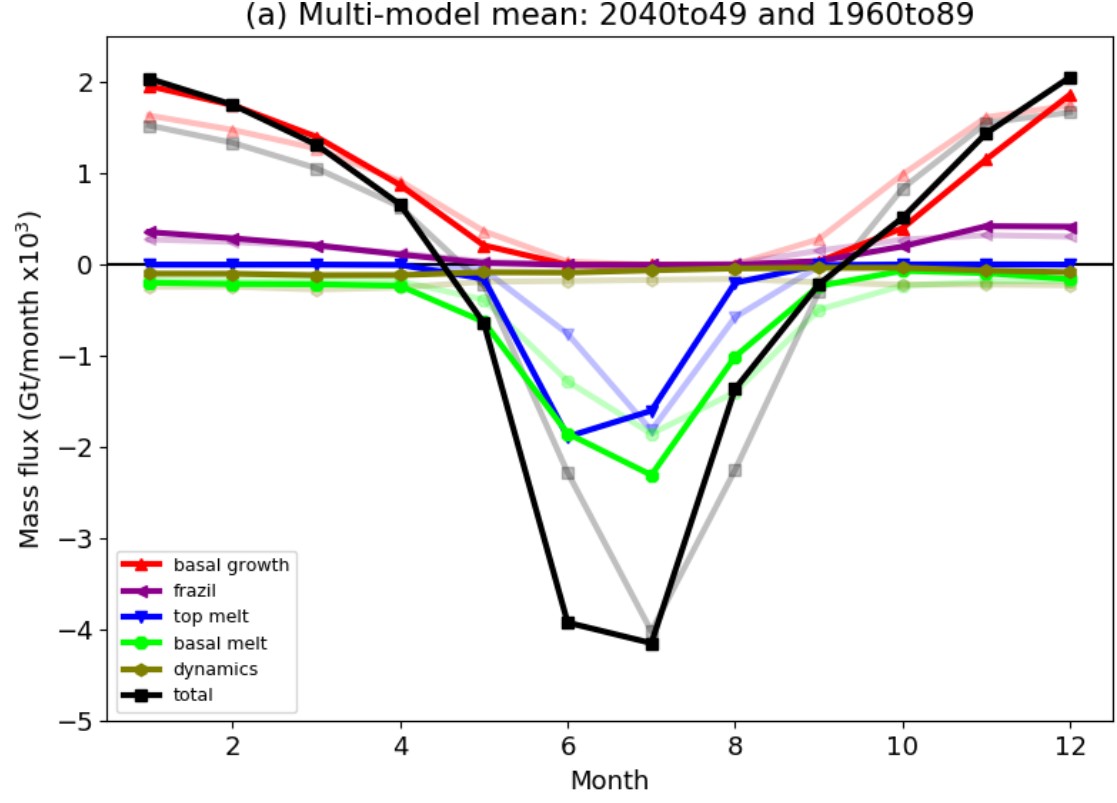

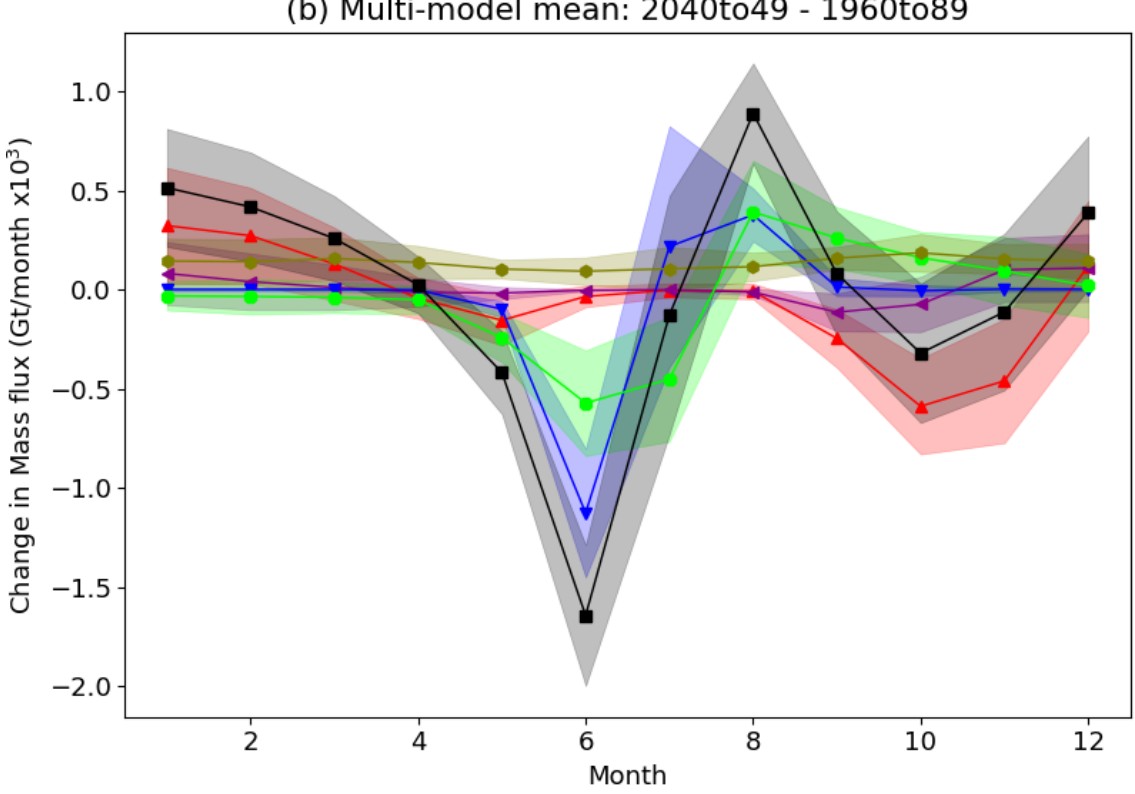

*Figure 11: Seasonal cycles of components of the Arctic sea ice mass budget for the multi-model mean. Values are summed over the region shown in Fig. 1. For each budget component, the multi-model mean is calculated by averaging the ensemble means for each model with data for that component. (a) Cycles for 1960-89 (faint) and 2040-49 (bold) (b) Difference (2040-49 − 1960-89).*

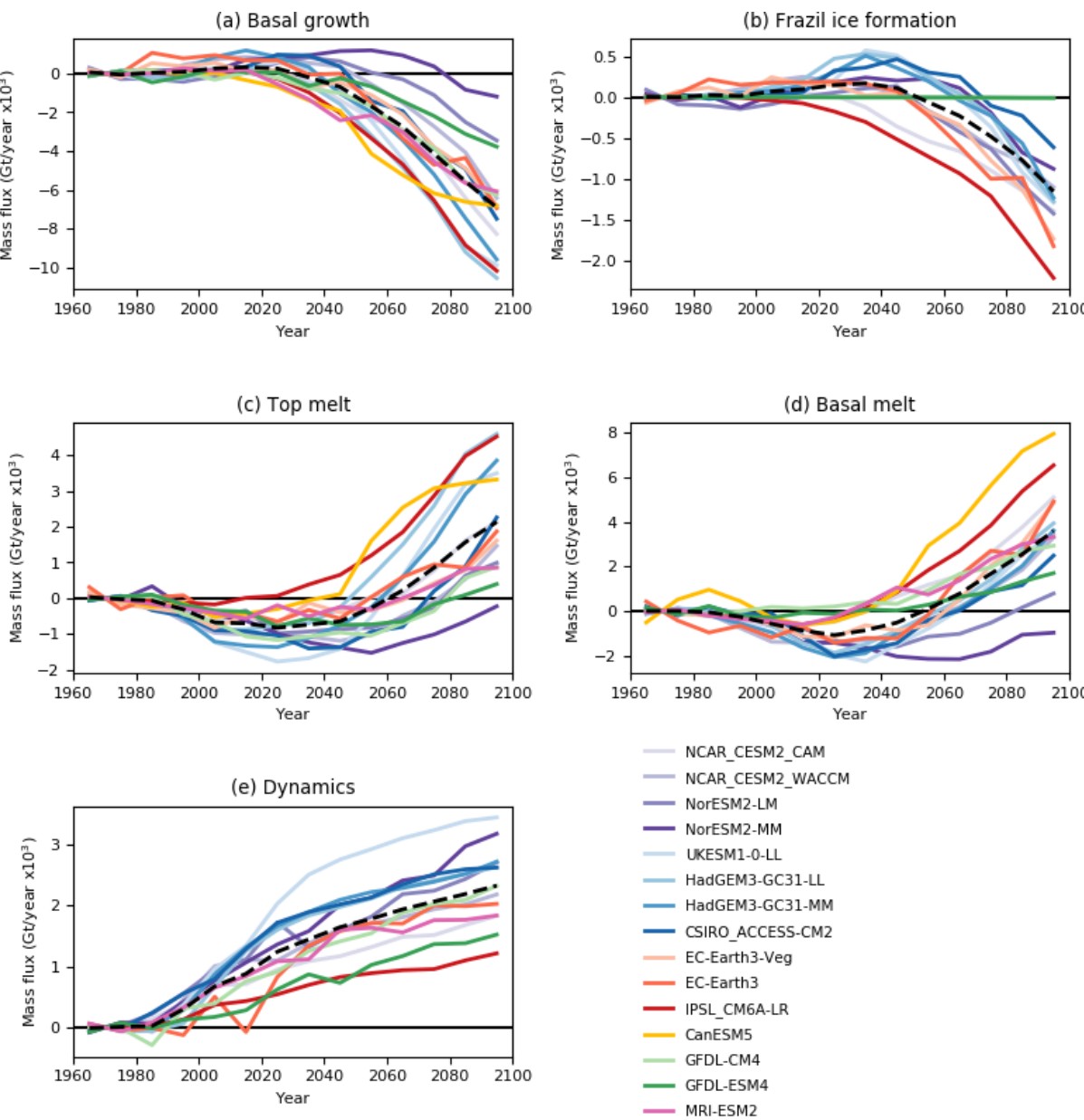


*Figure 12: Decadal mean anomalies (relative to the 1960-89) of the main components of the Arctic sea ice mass budget for all the models. Values are summed over the region shown in Fig. 1. The units are Gtx10³ year⁻¹, and the dashed lines on each plot show the multi-model mean.*


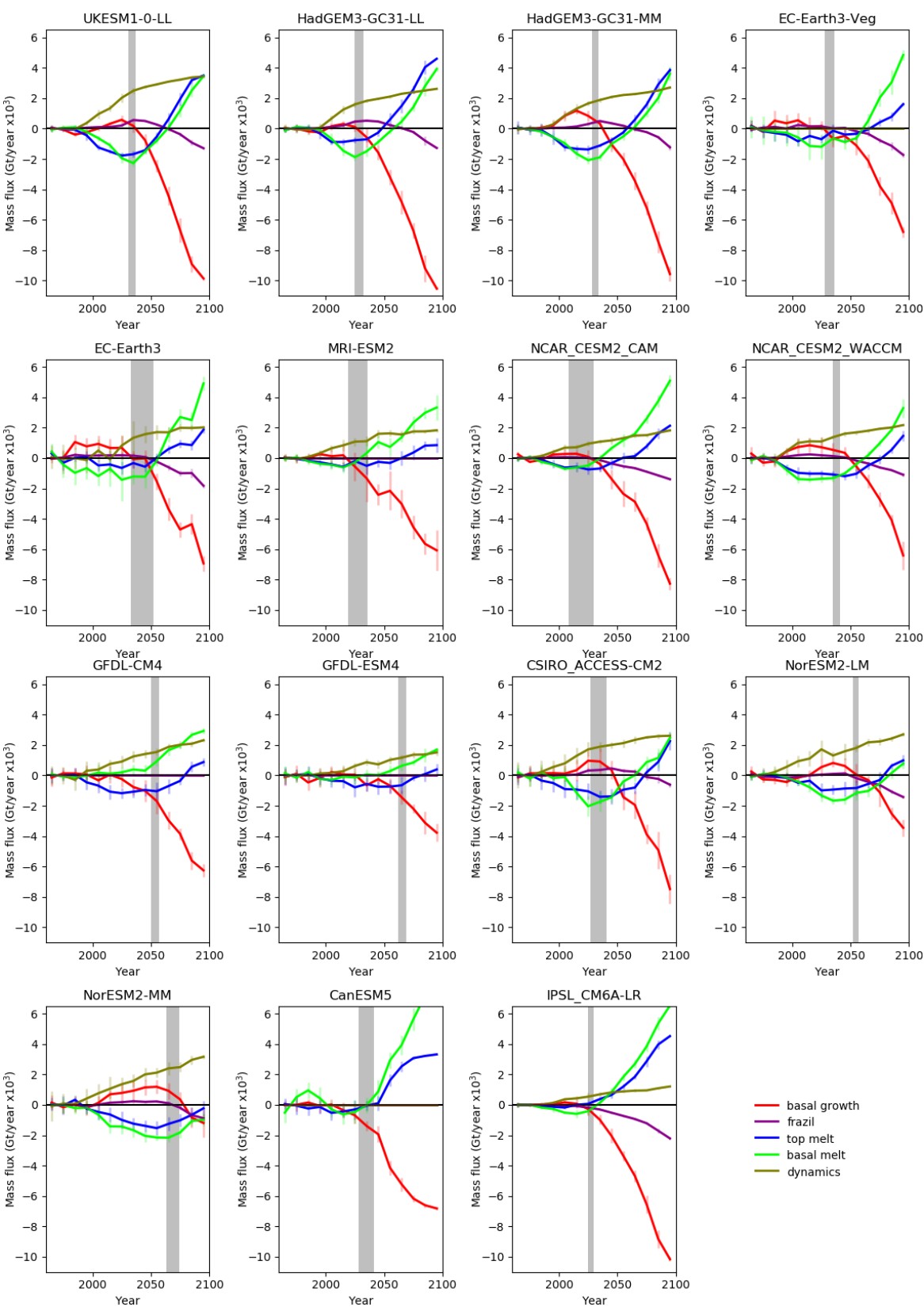

Figure 13: *Evolution of decadal mean components of the main terms in the Arctic sea ice mass budget for each model (Gt x10$^3$ per decade) Values are summed over the region shown in Fig. 1 and are anomalies relative to the 1960-89 mean. The shading shows a period during which each model becomes ice-free by the end of the summer. The lower bound is the first year when there is <1.0 million km$^2$ of ice in September, and the upper bound shows when the running 10 year mean first becomes <1.0 million km$^2$.*

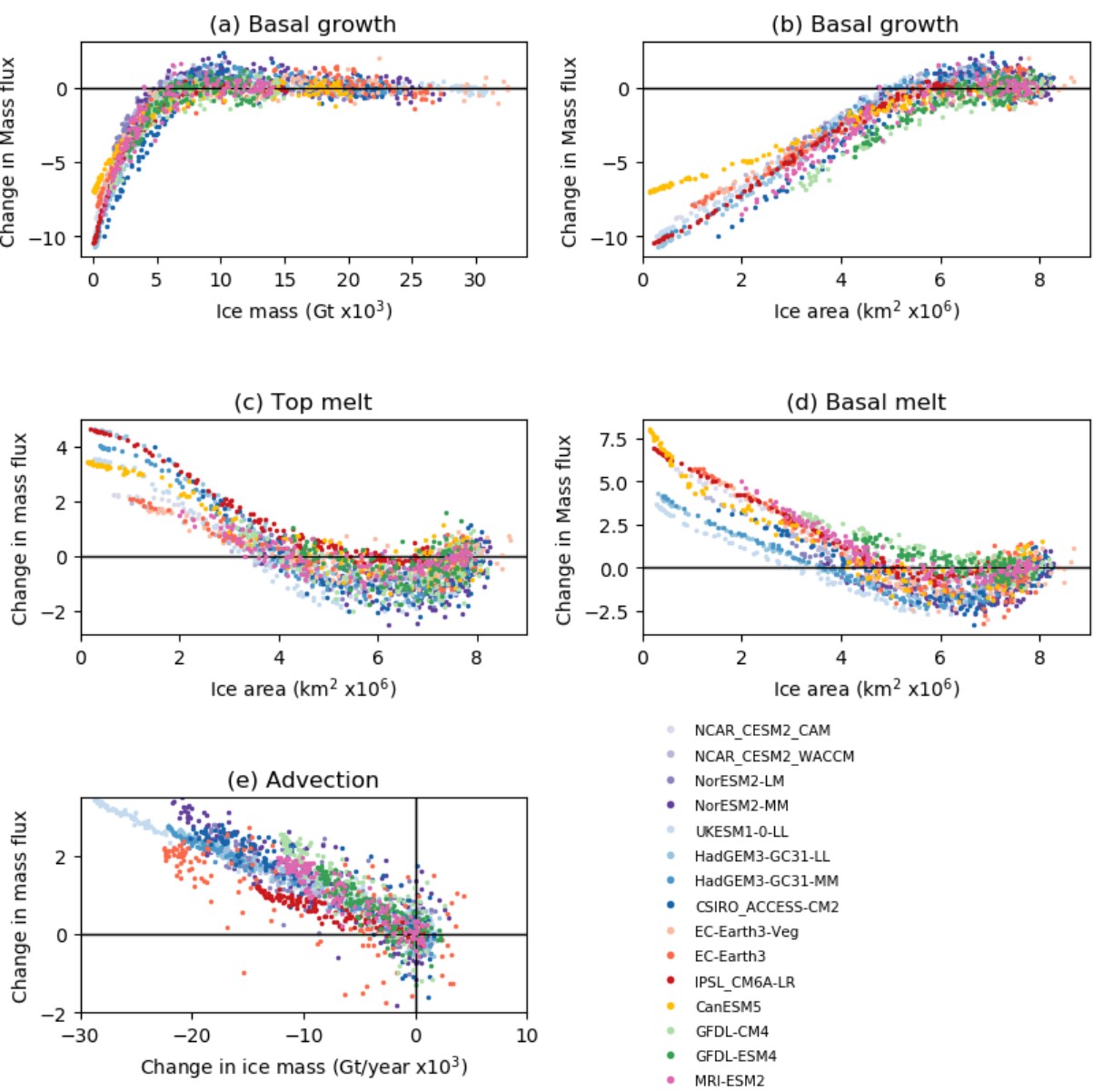

*Figure 14: Selected annual mean components of the Arctic sea ice mass budget for all the models,*
*plotted against ice state. All values are summed over the region shown in Fig. 1. The budget*
*components are anomalies relative to the 1960-89 mean, with units of Gt x10³ year⁻¹ (a) Basal*
*growth vs ice mass (b) Basal growth vs ice area (c) Top melt vs ice area (d) Basal melt vs ice area (e)*
*Advection vs change in ice mass (relative to 1960-89).*
