# Peer review of "An inter-comparison of the mass budget of the Arctic sea ice in CMIP6 models"

_The Cryosphere, 2019_

## Referee Comment (RC1) · Anonymous Referee #1 · 27 Mar 2020

General Comments

This paper is a first look at the Arctic ice mass budget terms saved by participating CMIP-6 modeling groups. The seasonal cycle for these mass budget terms during a reference period (1990-2009) is examined and differences between models discussed. Historical simulations concatenated by SPS5-85 projections are analyzed to examine differences in the evolution of total extent/mass and budget terms from 1960 through the end of the 21st century. The core finding is that, despite substantially different seasonal cycles and time trajectories, the differences in individual mass budget terms between models are relatively small. The paper is well written and constitutes a substantial effort in coordination between many different groups to provide output that can be easily compared. This effort is laudable and the results should provide a useful

reference as presented. As a reviewer, my job is to check the paper on originality, scientific quality, impact, and presentation quality. I think the paper is good to excellent in these categories but could gain some more impact through more focused analysis and presentation. As an author I am typically annoyed when reviewers ask me to write a paper I didn't mean to write so please take the following as suggestions. While a useful compilation of results as presented, the paper tends to scratch only the surface on some questions and falls a bit short of a deeper exploration. For example, a discussion of how the relatively small differences in the budget terms lead to rather large different mean states and trajectories would be interesting. This might perhaps be achieved by looking at cumulative budget differences (or anomalies from ensemble mean) so that one could see the relative contribution of budget terms to the state trajectory? Some of the inter-model differences seem to be obscured by the way they are presented. For example, some inter-model differences could perhaps be highlighted if shown as the change between periods (reference to late 21st century) instead of showing the full transition (Fig 4, 11). Maybe highlighting individual models as extremes might be useful to tease out some fundamental differences between models (or groups of models)? For example, the NCAR-CESM2 and CanESM-5 seem to have rather dramatic differences in their summer and winter trajectories, with the NCAR-CESM2 showing a very early decline in the summer and CanESM-5 being an outlier in its winter trajectory. This seems to be in part related to the very strong basal melt in CanESM-5? Figure 11, looking at mean state vs. flux terms, seems to indicate some clustering of models that may allow drawing conclusions about differences between models or groups. Some of the graphics could probably be condensed to highlight differences and focus on a particular thesis. The "forced" model run section could use some work to better relate it to the coupled runs and allow a sharpening of the conclusion beyond " both model sea-ice physics and atmospheric forcings are important". If this is too difficult maybe it should be cut entirely.

I hope you will find the below suggestions useful and recognize that they may not necessarily work or be too difficult to implement (or even make sense).

Specific Comments:

Abstract: Maybe a note about the lack of fundamental diversity in model physics would be useful here as a caveat. Line 81: Keen and Blockley (2018). Is it worth to add a section on whether or not any of the results presented here are different or the same as in your previous paper? Maybe a sentence in the discussion or conclusion? Line 110: Can you explain why "basal growth" and "basal melt" aren't the same quantity just with opposite sign? Line 121:"SSP5-8.5 scenario" ... given the debate about the likelihood of that scenario it might be useful to caveat that using an extreme scenario may actually be useful to highlight differences in the budget terms. Line 137: Why are both HadISST versions used? Is this supposed to provide some measure of the uncertainty in the ice concentration data? Uncertainty for PIOMAS might also be quantified in this context. Line 163.. "at least one of the observational data set"... see above comment about uncertainty in the observations. Line 176: "All models have their seasonal maximum ice mass in May" consistent with PIOMAS"... This must be due to the choice of domain since the full PIOMAS domain has its maximum volume in April not in May. Some note, either here or earlier on about how the domain can affect some of the variables though not likely the general conclusions, should be made. Figure 3: I think it would be better to show observations with some uncertainty measure (relates to above). PIOMAS uncertainty could be scaled from values given by Schweiger et al. 2011 to the domain? Why is the summer total ice mass trajectory not shown in 3b? Can you comment on the large differences in the summer/winter trajectories for NCAR-CESM2 vs. CanESM5? NCAR-CESM2 seems to be losing its ice much more rapidly in summer while CanESM-5 loses its winter ice much more rapidly than the other models? Line 199: "whereas other models show a more uniform decline". Don't you think this is likely because of the 2100 cut off and if you'd looked at beyond 2100 you'd see a similar flattening towards the end? Line 208. Figure 4b. Maybe it would be more informative to plot delta sea ice/delta temperature (sea ice sensitivity) between two periods. That would highlight model differences better. Line 211: "1960-1986" Maybe a statement why the 1960 through 1986 reference" period is chosen is different from the period over which Figure

2 seasonal cycle values are averaged (1990-2009). I suppose you were looking for greatest overlap with observations for the Figure 2 comparison. Line 230, figure 3b.. "20%" there is no percentage scale given that this number could be related to. Can you add a percentage axis in 3b or rephrase to one doesn't go looking for it. Fig 6,7,8 I find it pretty difficult to see the differences between models as plotted and the information density pretty low. Plotting seasonal cycles/trajectories relative to a multi-model mean would probably highlight differences better. Absolute values as shown in Figure 6, if deemed relevant, might be put into a table? I am looking for some visuals that highlight the conclusions drawn regarding each of the subsequent paragraphs. The reader has to do quite a bit of visual hunting to relate the statements made to the graphics. Line 292. . . "there is considerable variability between the models" again, I think removing the multi-model mean and showing this as anomalies relative to that mean would highlight the differences.

Line 309: "how the ice state impacts the evolution of each budget term". As said in the general comments, a discussion of how the budget terms accumulate to the evolution of the state would be useful. Line 314.. move (Fig 9a) to just after "area of ice" , maybe add a references (Bitz and Roe 2004) to link to think ice growth feedback process. Line 317: What does "in-situ" mean in this context? That sounds a bit off. Figure 9b: Heading says "Surface Melt" elsewhere named "Top Melt"? Fig 9/10 b/c. A little bit more discussion why the trajectories for surface and basal melt go in opposite direction for per unit area vs. total (9 bc/10bc) would be helpful. Something that states that while the melt at the base and surface of the remaining ice increases the total area of ice decreases, so there is less ice to which this process applies. I know you say this in Line 318 but maybe this could be made a bit clearer. Pointing to Figure 10, and maybe discussing the extreme case of CanESM5 would make this clear right away. I was scratching my head for a bit until this sunk in (maybe me). Fig 11. Wondering if this figure wouldn't be more informative if not every point of the trajectory were plotted for each model but rather the change from reference period to some target period (end of 21st) or before. This would probably highlight the model differences better. Line 385

Mass budgets for forced runs. Wondering if this shouldn't be a separate section in the body of the paper rather than in the discussion. I am struggling a bit what to make of this set of experiments and what the fundamental results are. I understand the goal of trying to quantify the relative impact of ice physics vs. forcing on the mass budget. I think the experiments with the same forcing but different physics accomplishes this w.r.t to one a couple of parameters. To make this meaningful though, I think this limited sample of sensitivities would somehow need be related to the spectrum of possible parameters in the CMIP-6 models. Does it represent an extreme and therefore establish a bracket of sea ice physics sensitivities? Similarly, the forcing sensitivities need to somehow be related to the range of atmospheric variables. What is the range between CORE II, DFS5.2 and MetUM coupled (whatever that is?). Of course that's not easy thing to quantify because the forcing consists of multiple variables and there could be compensating differences, but maybe temperature and downwelling radiation would capture a sufficient measure of the range. Of course this would have to be done for the CMIP-6 models then and that could get messy because of the coupling. I don't know what the solution is but I think as presented, this set of experiment probably confuses more than it adds to the discussion.

---

## Referee Comment (RC2) · Till Wagner (Referee) · 1 Jun 2020

In this manuscript, the authors compare the individual terms of the Arctic sea ice mass budget (predominantly surface and bottom melt, basal growth and frazil ice formation, and advection) from 14 CMIP6 models. This type of in-depth analysis had not been possible for previous CMIPs, as individual mass budget terms were not routinely reported.

The paper is very well written and structured, clearly illustrated, and the subject matter is a good fit for The Cryosphere. It presents an interesting result in that ∼half of the simulated annual ice loss is due to basal melt, and ∼one quarter each due to surface melt and advection out of the Arctic. Another central result is that ice formation oc-

curs predominantly as basal growth, with frazil ice formation playing a substantial role depending on a model's minimum frazil ice thickness. Finally, it is striking to see how consistent the partitioning of the individual mass budget terms is between models.

In light of this I recommend the paper for publication after minor revisions.

I agree with the other reviewer with regard to two general comments:

1) I found the paper could do with some more focus, and investigate model differences in more detail. In particular highlighting which differences are due to different physics (e.g., meltpond or radiation schemes) and which are due to different parameter values (such as minimum frazil ice thickness) would be of interest.

This brings me to my main comment. Many of the models share the same sea ice model components (CICE, LIM, SIS) and I believe it would be of interest what the differences are between models with the same sea ice component vs between models with different sea ice components. In the case this is not insightful, it would nevertheless be helpful to discuss the role of having shared sea ice components or not (e.g., for a reader like me who is keenly interested but no expert in the differences between sea ice model components).

2) In line with the other reviewer's comment, I found the section on the forced model runs somewhat vague and only tangentially related to central message of the paper. I would also suggest that this section be either incorporated more carefully or cut (which would further help focusing the main story).

Specific Comments:

l.65 "uncertainty" (not uncertainly)

l.68 " models' " (not model's)

l.74 maybe "emerging consensus" rather than "increasing appreciation"

l.135 what time periods are covered by the 3 observational products?

l.263 Is this a linear relationship? Would it be insightful to plot "% of frazil ice formation" vs "min frazil ice thickness"

l.309 period at end of sentence

l.350 "amount" (not about)

Fig 4 would it make sense to add the observations?

Fig 5b add legend

Fig 6 It is difficult to see the differences between the models. Is there a more concise way to present this data? In line with my major comment above, maybe it would be worth grouping the models by sea ice model component? e.g., instead of the bar plots have one subplot for all CICE models, one for all LIM models, etc, with each model's basal growth value indicated by a marker and the same for the other terms?

Fig 7 As for Fig 6 it is difficult to see differences between models.

Fig 9 and 10 are cropped on the left edge.

Fig 9 doesn't show units on vertical axis, Fig 10 does.

Fig 9b is "surface melt", Fig 10b is "top melt".

Fig 9a is "basal growth", Fig 10a is "frazil ice formation". Why are the different quantities shown, why not all 4 main terms for each Figure?

Fig 11 are the units on the vertical axis Gt year^-1 rather than kg year^-1?

Fig 12 maybe leave out the lines for lateral melt, snowice, evapsubl, since they are mostly negligible and make the figure harder to read?

Till Wagner

---

## Author Comment (AC1) · 11 Dec 2020

**General Comments**

This paper is a first look at the Arctic ice mass budget terms saved by participatingCMIP-6 modeling groups. The seasonal cycle for these mass budget terms during a reference period (1990-2009) is examined and differences between models discussed. Historical simulations concatenated by SPS5-85 projections are analyzed to examine differences in the evolution of total extent/mass and budget terms from 1960 through the end of the 21st century. The core finding is that, despite substantially different seasonal cycles and time trajectories, the differences in individual mass budget terms between models are relatively small. The paper is well written and constitutes a substantial effort in coordination between many different groups to provide output that can be easily compared. This effort is laudable and the results should provide a useful reference as presented. As a reviewer, my job is to check the paper on originality, scientific quality, impact, and presentation quality. I think the paper is good to excellent in these categories but could gain some more impact through more focused analysis and presentation. As an author I am typically annoyed when reviewers ask me to write a paper I didn't mean to write so please take the following as suggestions.

While a useful compilation of results as presented, the paper tends to scratch only the surface on some questions and falls a bit short of a deeper exploration. For example, a discussion of how the relatively small differences in the budget terms lead to rather large different mean states and trajectories would be interesting. This might perhaps be achieved by looking at cumulative budget differences (or anomalies from ensemble mean) so that one could see the relative contribution of budget terms to the state trajectory?

Thank you for these suggestions. We have updated the discussion of how the model budgets change during the 21$^{st}$ century, and now include plots showing the evolution of the anomalies in the budget terms, both for the multi-model mean (Fig. 10b) and the

individual models (Fig. 13). Hopefully these plots show more clearly the relative changes in the different budget terms, and how the balance of these alters as the model integrations progress. This is also discussed in the updated text.

Some of the inter-model differences seem to be obscured by the way they are presented. For example, some inter-model differences could perhaps be highlighted if shown as the change between periods (reference to late 21st century) instead of showing the full transition (Fig 4, 11).

Fig 4 (Fig 3 in the revised manuscript) is now updated to show only the first panel, and numbers quantifying the changes for two periods relative to 1960-89 are given in the text. We have retained the evolution over the entire period (relative to 1960-89) in the figure as it illustrates the divergence in response between the different models as the 21st century progresses.

For Fig 11 (Fig 14 in the revised manuscript), the aim of this figure is to highlight the underlying similarity in the evolution of the model responses, which is lost if two time periods are chosen as the model budgets change at different rates. We have updated the text in this section, so that hopefully this is more clearly explained.

Maybe highlighting individual models as extremes might be useful to tease out some fundamental differences between models (or groups of models)? For example, the NCAR-CESM2 and CanESM-5 seem to have rather dramatic differences in their summer and winter trajectories, with the NCAR-CESM2 showing a very early decline in the summer and CanESM-5 being an outlier in its winter trajectory. This seems to be in part related to the very strong basal melt in CanESM-5?

Thank you for this suggestion. We were not able to use the CanESM5 model for this purpose, as some of the main budget terms were unavailable for this model (the frazil and advection components). However, in section 6 we do now highlight models with different ice state trajectories as we discuss the differences between the models.

Figure 11, looking at mean state vs. flux terms, seems to indicate some clustering of models that may allow drawing conclusions about differences between models or groups.

While we have not specifically used figure 11 for this, we have been able to identify differences between the model budgets during the reference period that are consistent with some of the sea ice parametrization choices. This is explored in the new section 5.

Some of the graphics could probably be condensed to highlight differences and focus on a particular thesis.

We agree and have updated a number of the figures to better illustrate the points made in the manuscript. For example, figure 7 (figure 6 in the updated manuscript) now has a pane for each budget term rather than for each model, to better illustrate the differences between the models. Figures 8,9 and 10 (replaced by new figures 10 and 12 in the updated manuscript) now show decadal means rather than annual means, to show the evolution and differences between the models more clearly. There are further details in the specific responses below.

The "forced" model run section could use some work to better relate it to the coupled runs and allow a sharpening of the conclusion beyond " both model sea-ice physics and atmospheric forcings are important". If this is too difficult maybe it should be cut entirely.

We have now included a new section 'Understanding differences between the CMIP6 models' (Section 5), where the forced experiments are introduced and used to help understand differences between the model mass budgets during the reference period.

I hope you will find the below suggestions useful and recognize that they may not necessarily work or be too difficult to implement (or even make sense).

**Specific Comments**

Abstract: Maybe a note about the lack of fundamental diversity in model physics would be useful here as a caveat.

This has now been added.

Line 81: Keen and Blockley (2018). Is it worth to add a section on whether or not any of the results presented here are different or the same as in your previous paper? Maybe a sentence in the discussion or conclusion?

We have now added some text in the discussion to mention that these results are broadly consistent with the findings of Keen and Blockley (2018).

Line 110: Can you explain why "basal growth" and "basal melt" aren't the same quantity just with opposite sign?

These two terms are defined separately in the SIMIP mass budget and are important at different times of the year, hence we feel it is important to keep them separate in the analysis. The factors affecting each can also be different.

Line 121:"SSP5-8.5 scenario"...given the debate about the likelihood of that scenario it might be useful to caveat that using an extreme scenario may actually be useful to highlight differences in the budget terms.

We agree and have added some text to mention this.

Line 137: Why are both HadISST versions used? Is this supposed to provide some measure of the uncertainty in the ice concentration data? Uncertainty for PIOMAS might also be quantified in this context.

Yes, that's right. We have clarified this in the text and added a comment about PIOMAS uncertainty.

Line 163.. "at least one of the observational data set"...see above comment about uncertainty in the observations.

This sentence has been re-worded.

Line 176: "All models have their seasonal maximum ice mass in May" consistent with PIOMAS"...This must be due to the choice of domain since the full PIOMAS domain

has its maximum volume in April not in May. Some note, either here or earlier on about how the domain can affect some of the variables though not likely the general conclusions, should be made.

This has been noted in section 2, where the domain is first mentioned.

Figure 3: I think it would be better to show observations with some uncertainty measure (relates to above). PIOMAS uncertainty could be scaled from values given by Schweiger et al. 2011 to the domain? Why is the summer total ice mass trajectory not shown in 3b? Can you comment on the large differences in the summer/winter trajectories for NCAR-CESM2 vs. CanESM5? NCAR-CESM2 seems to be losing its ice much more rapidly in summer while CanESM-5 loses its winter ice much more rapidly than the other models?

An uncertainty range for the observations has been added to each plot.

We did not include both summer and winter mass trajectories in Fig 3b (Fig. 2b in the revised manuscript) as the summer trajectories did not add any significant extra information: relative differences in the rate of mass decline between the different models are very similar whether looking at the March or September values. We now mention this in the text. In addition, the figure would have become quite cluttered as there is not a clear separation between the summer and winter values.
We have added text to highlight the fact that models with the fastest decline in summer ice area are not necessarily those with the fastest winter decline, although we have not identified a particular model feature that may cause this.

Line 199: "whereas other models show a more uniform decline". Don't you think this is likely because of the 2100 cut off and if you'd looked at beyond 2100 you'd see a similar flattening towards the end?

Yes, we agree and have updated the text accordingly.

Line 208. Figure 4b. Maybe it would be more informative to plot delta seaice/delta temperature (sea ice sensitivity) between two periods. That would highlight model differences better.

Thank you for this suggestion. We did generate a plot of the ice sensitivity, which we planned to include. However, as the sensitivity of the CMIP6 models is now discussed in the recently published SIMIP community paper we decided to refer to this instead and have updated the appropriate text accordingly.
(SIMIP Community (2020). Arctic sea ice in CMIP6. Geophysical ResearchLetters,47, e2019GL086749. https://doi.org/10.1029/2019GL086749)

Line 211: "1960-1986" Maybe a statement why the 1960 through 1986 reference" period is chosen is different from the period over which Figure 2 seasonal cycle values are averaged (1990-2009). I suppose you were looking for greatest overlap with observations for the Figure 2 comparison.

Yes, that's right. For the budget reference period we chose a period during which the ice area and mass are relatively stable. For the seasonal cycles of ice state, we chose a later period with more observational data (including satellite data). This has been mentioned in the text.

Line 230, figure 3b.."20%" there is no percentage scale given that this number could be related to. Can you add a percentage axis in 3b or rephrase to one doesn't go looking for it.

I think this refers to figure 5b (figure 4b in the updated manuscript), and the percentages have now been added to the figure.

Fig 6,7,8 I find it pretty difficult to see the differences between models as plotted and the information density pretty low. Plotting seasonal cycles/trajectories relative to a multi-model mean would probably highlight differences better. Absolute values as shown in Figure 6, if deemed relevant, might be put into a table? I am looking for some visuals that highlight the conclusions drawn regarding each of the subsequent paragraphs. The reader has to do quite a bit of visual hunting to relate the statements made to the graphics.

Figure 6 (figure 5 in the revised manuscript) has been updated to show only the main budget terms, and to also include the multi-model mean for reference. Figure 7 (figure 6 in the revised manuscript) has been altered so that the seasonal cycles of all the models (for each budget component) are on the same plot. Figure 8 (Figure 13 in the revised manuscript) has been updated to show anomalies relative to the reference period, and decadal means rather than annual means. We feel that these are an improvement over the original versions.
We tried generating versions of both Figure 6 and Figure 7 using anomalies relative to the multi-model mean but found these to be hard to interpret. In addition, one of the aims of figure 6 (figure 5 in the revised manuscript) is to show the broad similarity of the overall balance of the model budget terms, which was lost when plotting the anomalies.

Line292. . ."there is considerable variability between the models" again, I think removing the multi-model mean and showing this as anomalies relative to that mean would highlight the differences.

The new fig. 6e, while still showing the absolute values, does now show the differences between the different models more clearly as the vertical scale is more appropriate.

Line 309: "how the ice state impacts the evolution of each budget term". As said in the general comments, a discussion of how the budget terms accumulate to the evolution of the state would be useful.

We have revised the discussion of the evolution of the ice budget, and the associated figures. In particular, we now show how the anomalies in the main budget terms evolve, both for the multi-model mean and for the individual models, and that different terms are important for the ice loss at different stages in the ice decline.

Line 314.. move (Fig 9a) to just after "area of ice" , maybe add a references (Bitz and Roe 2004) to link to think ice growth feedback process.

We have added this reference – thank you for mentioning it.

Line317: What does "in-situ" mean in this context? That sounds a bit off. Figure 9b: Heading says "Surface Melt" elsewhere named "Top Melt"?

The figure heading has been corrected, and the text modified. We meant the values expressed 'per unit area of the ice', and this has been re-worded in the revised version of this section so that it is clearer.

Fig 9/10 b/c. A little bit more discussion why the trajectories for surface and basal melt go in opposite direction for per unit area vs. total (9 bc/10bc) would be helpful. Something that states that while the melt at the base and surface of the remaining ice increases the total area of ice decreases, so there is less ice to which this process applies. I know you say this in Line 318 but maybe this could be made a bit clearer. Pointing to Figure 10, and maybe discussing the extreme case of CanESM5 would make this clear right away. I was scratching my head for a bit until this sunk in (maybe me).

The discussion of these trajectories been updated and is now illustrated by the multi-model mean (new figure 10). We have improved the text so that hopefully this is clearer, for example the summary to this section contains the following:
"… the amount of basal melt per unit area of the ice continues to increase throughout the 21$^{st}$ century (Fig. 10c), but the total amount of ice lost by basal melt reaches a maximum in the 2020s and declines after that because there is a reduced area of ice over which the melting can occur."

Fig 11. Wondering if this figure wouldn't be more informative if not every point of the trajectory were plotted foreach model but rather the change from reference period to some target period (end of 21st) or before. This would probably highlight the model differences better.

As mentioned above, we have retained Fig 11 (Fig 14 in the updated manuscript), as the aim of this figure is to highlight the underlying similarity in the evolution of the model responses, which is lost if two time periods are chosen as the model budgets change at different rates. We have updated the text in this section, so hopefully this is more clearly explained.

Line 385 Mass budgets for forced runs. Wondering if this shouldn't be a separate section in the body of the paper rather than in the discussion. I am struggling a bit what to make of this set of experiments and what the fundamental results are. I understand the goal of trying to quantify the relative impact of ice physics vs. forcing on the mass budget. I think the experiments with the same forcing but different physics accomplishes this w.r.t to one a couple of parameters. To make this meaningful though, I think this limited sample of sensitivities would somehow need be related to the spectrum of possible parameters in the CMIP-6 models. Does it represent an extreme and therefore establish a bracket of sea ice physics sensitivities? Similarly, the forcing sensitivities need to somehow be related to the range of atmospheric variables. What is the range between CORE II, DFS5.2 and MetUM coupled (whatever that is?). Of course that's not easy thing to quantify because the forcing consists of multiple variables and there could be compensating differences, but maybe temperature and downwelling radiation would capture a sufficient measure of the range. Of course this would have to be done for theCMIP-6 models then and that could get messy because of the coupling. I don't know what the solution is but I think as presented, this set of experiment probably confuses more than it adds to the discussion.

We have now added a new section to the paper ('Understanding differences between the models'), where the forced runs are introduced and used to help understand the differences in the model budgets. We have added some comments about the

sensitivities considered in the forced models, and how these relate to the range of options covered in the CMIP6 models (for example they do not represent an extreme), but as mentioned it is difficult to quantify this. We have been able to identify some common features in the model budgets that are consistent with some of the parametrization choices.

---

## Author Comment (AC2) · 11 Dec 2020

Till Wagner (Referee)

In this manuscript, the authors compare the individual terms of the Arctic sea ice mass budget (predominantly surface and bottom melt, basal growth and frazil ice formation, and advection) from 14 CMIP6 models. This type of in-depth analysis had not been possible for previous CMIPs, as individual mass budget terms were not routinely reported.

The paper is very well written and structured, clearly illustrated, and the subject matter is a good fit for The Cryosphere. It presents an interesting result in that ~half of the simulated annual ice loss is due to basal melt, and ~one quarter each due to surface melt and advection out of the Arctic. Another central result is that ice formation occurs predominantly as basal growth, with frazil ice formation playing a substantial role depending on a model's minimum frazil ice thickness. Finally, it is striking to see how consistent the partitioning of the individual mass budget terms is between models.

In light of this I recommend the paper for publication after minor revisions.

I agree with the other reviewer with regard to two general comments:

1) I found the paper could do with some more focus, and investigate model differences in more detail. In particular highlighting which differences are due to different physics (e.g., meltpond or radiation schemes) and which are due to different parameter values (such as minimum frazil ice thickness) would be of interest.

This brings me to my main comment. Many of the models share the same sea ice model components (CICE, LIM, SIS) and I believe it would be of interest what the differences are between models with the same sea ice component vs between models with different sea ice components. In the case this is not insightful, it would nevertheless be helpful to discuss th role of having shared sea ice components or not (e.g., for a reader like me who is keenly interested but no expert in the differences between sea ice model components).

2) In line with the other reviewer's comment, I found the section on the forced model runs somewhat vague and only tangentially related to central message of the paper. I would also suggest that this section be either incorporated more carefully or cut (which would further help focusing the main story).

In response to both your general comments (and those of the other reviewer), we have now included a new section 'Understanding differences between the CMIP6 models' (Section 5), where the forced experiments are introduced and used to help understand differences between the model mass budgets during the reference period. We have also looked more closely at the model budget differences and identified links between some of the budget terms and the formulation of the sea ice component.

Specific Comments:

l.65 "uncertainty" (not uncertainly)
l.68 " models' " (not model's)
l.74 maybe "emerging consensus" rather than "increasing appreciation"

These changes have been made.

l.135 what time periods are covered by the 3 observational products?

For the seasonal cycles shown in figure 2 (figure 1 in the revised manuscript), we use the years 1990-2009 for each observational dataset.
For the trajectories shown in figures 3 and 4 (figures 2 and 3 in the revised manuscript), we now use the following time periods:
- OSI-SAF, HadISST1.2 and HadISST2,2,0,0: 1979-2015
- PIOMAS: 1979-2018
- HadCRUT4: 1960-2019
This is now clarified in the manuscript.

l.263 Is this a linear relationship? Would it be insightful to plot "% of frazil ice formation" vs "min frazil ice thickness"
That is an interesting question. There is some evidence that it may be a linear relationship (see plot below). However, since we submitted this work the term representing frazil ice formation in the MRI-ESM2 model has been added to the basal growth term, which means we no longer have a frazil ice term for a model with a minimum frazil thickness of 20cm. So as there are not many data points we have not focussed on this in the text.

[Figure]

*Plot of frazil ice formation (as a percentage of total ice growth) for the reference period 1960 to 89 for each model, plotted against the minimum thickness of frazil ice formation.*

l.309 period at end of sentence
l.350 "amount" (not about)
The test in this section has now been updated, and these mistakes corrected.

Fig 4 would it make sense to add the observations?
Yes, we have now added observations to Figure 4a. (figure 3 in the revised manuscript). Figure 4b has been removed, as we instead reference the recently published SIMIP community paper: SIMIP Community (2020). Arctic seaice in CMIP6.Geophysical ResearchLetters,47, e2019GL086749. https://doi.org/10.1029/2019GL086749

Fig 5b add legend
A legend has been added. (figure 4b in the revised manuscript).

Fig 6 It is difficult to see the differences between the models. Is there a more concise way to present this data? In line with my major comment above, maybe it would be worth grouping the models by sea ice model component? e.g., instead of the bar plots have one subplot for all CICE models, one for all LIM models, etc, with each model's basal growth value indicated by a marker and the same for the other terms?
Fig 7 As for Fig 6 it is difficult to see differences between models.

Figures 6 and 7 have now been updated (figures 5 and 6 in the revised manuscript), and hopefully are now clearer. Thank you for the suggestion to group the models. A new figure 9 shows the main budget terms grouped according to a number of key model parametrizations and settings, and this has proved very useful in helping to understand some of the budget differences.

Fig 9 and 10 are cropped on the left edge.
Fig 9 doesn't show units on vertical axis, Fig 10 does.
Fig 9b is "surface melt", Fig 10b is "top melt".

These figures have been replaced in the updated manuscript. and Fig. 9 is replaced by the new Fig. 10c, which shows the multi-model mean values per unit area of the ice., and Fig 10 is replaced by the new Fig. 12. We have checked to ensure that the units are noted in each case, that the figures are not cropped, and that the headings are correct.

Fig 9a is "basal growth", Fig 10a is "frazil ice formation". Why are the different quantities shown, why not all 4 main terms for each Figure?

Yes, we realise now that this was rather confusing. The figures for this section have been altered, and the evolution of the budget terms is now shown for all the main terms (new Fig. 12). The values per unit area of the ice are now shown for the multi-model mean (new Fig. 10c), and only for those processes that occur over the ice surface. This is explained in the caption.

Fig 11 are the units on the vertical axis Gt yearˆ-1 rather than kg yearˆ-1?

Yes, thank you for spotting this. It has now been corrected (new Fig. 14)

Fig 12 maybe leave out the lines for lateral melt, snowice, evapsubl, since they are mostly negligible and make the figure harder to read?

Thank you for this suggestion. We have now removed these smaller terms from all the figures apart from figure 4 (where the budget is first introduced). This is a definite improvement.

---

## Author Response (AR2)

Dear Chris,

Thank you for your quick response to our revised manuscript, and we are pleased that you are happy with our responses to the reviews. Our responses to your comments are included below in blue, and the majority of these have now been incorporated into the revised manuscript.

In addition, we have altered the larger tables so that all the manuscript pages are now in portrait orientation, as requested at the file verification stage. I think the tables are rather harder to read as a result, but hopefully the final 'typeset' version will look better.

Thank you for your work as editor of our paper. We have appreciated your quick responses and your understanding of the delays that have occurred during this year.

Yours sincerely, on behalf of all the authors,
Ann Keen.

Dear Ann – the concerns of the reviewers are clearly addressed in the revised manuscript. The study provides a comprehensive assessment of CMIP6 model behaviour with respect to sea ice, and is a highly relevant contribution to The Cryosphere. I really like your additions to the manuscript, which now more clearly identify relationships between model formulation and ice budget terms. I have identified just a few minor remaining issues to address (note that line numbers refer to the clean version of the revised manuscript with no tracked changes). Thanks,
Chris Derksen

The level of detail with respect to model descriptions is inconsistent across the Appendix (e.g. very little information on CanESM). Can this be harmonized somewhat?

We have now added a little more information about CanESM, and shortened some of longer model sections so that they are of a more consistent length and level of detail.

Line 124: consider changing to: "Snow ice: ice formation due to the transformation of snow to sea ice due to surface flooding"

This has been changed as suggested.

Line 287: "If we consider the total amount of winter ice growth (here taken as the sum of the frazil and basal growth terms), the spread in modelled values is 3.9x103Gt, compared to the larger range of 5.9x103 Gt for the basal growth alone." Can you provide a brief explanation for the apparent compensatory behavior for frazil formation versus basal ice growth?

The following text has been added: The lower the value of the minimum frazil ice thickness, the more quickly the frazil ice growth can transition to basal growth.

Line 309: can you remove the phrase "likely to be"?

This has been removed.

Line 341: "a simple scheme to account for the loss of drifting snow". I think a reference should be added here to Lecomte, O., Fichefet, T., Flocco, D., Schröder, D., and Vancoppenolle, M.: Impacts of wind-blown snow redistribution on melt ponds in a coupled ocean – sea ice model, Ocean Model, 87, 67–80, https://doi.org/10.1016/j.ocemod.2014.12.003, 2014.

Actually, it not the Lecomte et al. 2014 snow redistribution scheme, but the one described in Schroeder et al. (2019), which is referenced at the end of the sentence.

Line 350: I don't see a description anywhere of DFS versus CORE forcing?

We now include the following text: Replacing the CORE forcing which is based on NCEP reanalysis (Large and Yeager, 2009), with DFS forcing based on ERA-interim reanalysis (Dussin et al, 2016), increases the top melt (Fig. 8c) and decreases the basal melt (Fig. 8a), resulting in higher sea ice area and mass.

Line 396: specify that "The remaining models have a minimum frazil ice thickness…"

This has been changed as suggested.

Line 449: consider changing the header "Dynamics" to "Ice Advection" for consistency.

This has been changed as suggested.

Line 556: perhaps add a statement that insight on ice advection from the projections is limited because analysis of winds and large-scale atmospheric circulation (and hence simulated ice motion) were out of scope for this study.

We have added the following sentence: Further insight on the changes in ice advection would require an analysis of winds and large-scale atmospheric circulation, and the associated ice motion, which is outside the scope of this study.

Line 591: "Overall, models with the largest decline in basal ice growth by the end of the 21st century tend to be those with the larger decline in winter ice cover." In Section 6.1 and Figure 10c, you discuss and show the changes also as a function of ice unit area. Is there value in including this perspective also in Figure 12 (as noted on line 621)?

We have added a reference to Figs. 2a and 12a at the end of the sentence mentioned. When revising the manuscript, we did consider retaining the 'per ice area' plots for all the models – essentially figure 9 in the original version of the manuscript, but for decadal rather than annual mean values, but in the end we did not feel that this figure added significantly to the discussion. Once we had generated the new fig 10c using the multi-model mean, this seemed to more clearly demonstrate the points we wanted to make.

Non-public comments to the Author:
Thanks for the revised manuscript, Ann. This is a very comprehensive study and an important analysis of CMIP6 models. I identified just some minor things to clean up…
Thanks for your contribution to The Cryosphere.
Chris